# Impact of atmospheric forcing uncertainties on Arctic and Antarctic sea ice simulations in CMIP6 OMIP

Xia Lin[1, 2], François Massonnet [1], Thierry Fichefet [1], Martin Vancoppenolle[3]

[1]Earth and Life Institute, Université catholique de Louvain, Louvain-la-Neuve, 1348, Belgium

[2]Southern Marine Science and Engineering Guangdong Laboratory (Zhuhai), Zhuhai, 519000, China

[3]Laboratoire d'Océanographie et du Climat, CNRS/IRD/MNHN, Sorbonne Université, 75252, Paris, France

*Correspondence to*: Xia Lin (xia.lin@uclouvain.be)

**Abstract.** Atmospheric reanalyses are valuable datasets to drive ocean-sea ice general circulation models and to propose multi-decadal reconstructions of the ocean-sea ice system in polar regions. However, these reanalyses exhibit biases in these regions. It was previously found that the representation of Arctic and Antarctic sea ice in models participating in the Ocean Model Intercomparison Project Phase 2 (OMIP2, using the Japanese 55-year atmospheric reanalysis) was significantly more realistic than in the OMIP1 (forced by atmospheric state from the Coordinated Ocean-ice Reference Experiments version 2, CORE-II). To understand why, we study the sea ice concentration budget and its relations to surface heat and momentum fluxes, as well as the connections between the simulated ice drift and the ice concentration, the ice thickness and the wind stress in a subset of three models (CMCC-CM2-SR5, MRI-ESM2-0, and NorESM2-LM). These three models are representative of the ensemble and are the only ones to provide the surface fluxes and the tendencies of ice concentration attributed to dynamic and thermodynamic processes required for the ice concentration budget analysis. The sea ice simulations of two other models (EC-Earth3 and MIROC6) forced by both CORE-II and JRA55-do reanalysis are also included in the analysis. It is found that negative summer biases in high-ice concentration regions and positive biases in the Canadian Arctic Archipelago (CAA) and central Weddell Sea (CWS) regions are reduced from OMIP1 to OMIP2 due to surface heat fluxes changes. Net shortwave radiation fluxes provide key improvements in the Arctic interior, CAA and CWS regions. There is also an influence of improved surface wind stress in OMIP2 giving better winter Antarctic ice concentration and the Arctic ice drift magnitude simulations near the ice edge. The ice velocity direction simulation in the Beaufort Gyre and the Pacific and Atlantic sectors of the Southern Ocean in OMIP2 are also improved owing to surface wind stress changes. This study provides clues on how improved atmospheric reanalysis products influence sea ice simulations. Our findings suggest that attention should be paid to the radiation fluxes and winds in atmospheric reanalyses in polar regions.

# 1 Introduction

Sea ice is an important component of the polar climate system. At high latitudes, the presence of sea ice affects the exchanges of heat, momentum and freshwater fluxes between the atmosphere and the ocean.
Sea ice has experienced dramatic changes during recent decades, especially in the Arctic where the total sea ice extent dramatically decreased over the satellite observing period (Comiso et al. 2008; Stroeve and Notz, 2018). In the Antarctic, the total sea ice extent increased slightly, but statistically significantly, to a record high in 2014 and decreased dramatically to the lowest value in 2017 over the satellite record (Parkinson, 2019; Fogt et al., 2022). A record low sea ice extent was set in 2022
(Raphael and Handcock, 2022). Sea ice variability can drive changes in the atmospheric energy budget and circulation (Krikken and Hazeleger, 2015; Smith et al., 2017, 2022), as well as surface fluxes into the ocean and ocean circulation (Haumann et al., 2016; Sévellec et al., 2017; Meneghello et al., 2018).

A good simulation of sea ice is crucial to improve model predictions and climate change projections.
Yet, limitations still exist in both fully coupled climate models and ocean-sea ice models. For the Arctic, the observed decline of sea ice cover lies within the spread of modeled trends, but the multi-model mean trend is underestimated in the third, fifth and sixth phases of the Coupled Model Intercomparison Project (CMIP, Stroeve et al. 2007; Massonnet et al. 2012; Rosenblum and Eisenman 2017; Notz and SIMIP Community, 2020). The observed accelerated ice drift speed is not captured in CMIP3 models
(Rampal et al., 2011), while the accelerated ice drift speed is produced in winter but not in summer in CMIP5 models (Tandon et al., 2018). Large ice edge and thickness errors in Arctic subregions are identified from the spatial distribution of sea ice in CMIP6 models (Stroeve et al. 2014; Watts et al., 2021). For the Antarctic, the CMIP5 and CMIP6 models fail to capture the slightly increase of observed ice extent from 1979 to 2015 and they do not properly simulate the mean state and interannual
variability of the ice cover (Mahlstein et al., 2013; Turner et al, 2013; Zunz et al. 2013; Shu et al., 2015, 2020; Roach et al., 2020). Large biases are also noticed in simulations conducted with ocean-sea ice models driven by atmospheric reanalysis data, in particular on the Antarctic sea ice extent variability and the ice thickness and motion in both hemispheres (e.g., Massonnet et al. 2011; Lecomte et al., 2016; Chevallier et al. 2017). By performing sensitivity experiments with these ocean-sea ice models, one can
gain some insight into the origins of those biases. The focus of the present study is to quantify and understand how the sea ice simulation can be improved by changing atmospheric forcing fields in ocean-sea ice models.

Atmospheric reanalyses are particularly valuable in polar regions where in-situ observations are scarce.
However, these reanalyses have their limitations and biases (e.g., Lindsay et al., 2014; Bromwich et al., 2016; Barthélemy et al., 2018; Lin et al., 2018). Previous studies have shown that differences in the atmospheric forcing fields can affect the ocean-sea ice model simulations of the Arctic monthly mean

sea ice thickness and total sea ice volume (e.g., Hunke and Holland, 2007; Lindsay et al., 2014; Sterlin et al., 2021), the Arctic and Antarctic sea ice concentration in the marginal ice zones (Chaudhuri et al.,

2016) as well as the Antarctic sea ice extent, motion and thickness (Barthélemy et al., 2018). Wu et al. (2020) also showed the positive impacts of high-frequency (hourly to daily) atmospheric fluctuations on the Antarctic sea ice simulation, which implies that driving an ocean-sea ice model with a reanalysis that is developed at enhanced temporal and spatial resolution can help capture the small scale atmospheric processes and eventually improve the representation of sea ice.


The CMIP6 Ocean Model Intercomparison Project (OMIP, Griffies et al., 2016) provides global ocean-sea ice model simulations in two streams of model experiments: OMIP1, forced by the Coordinated Ocean-ice Reference Experiments, version 2 interannual dataset (CORE-II; Large and Yeager, 2009), and OMIP2, forced by the updated Japanese 55-year atmospheric reanalysis (JRA55-do; Tsujino et al.,

2018). The design of the CMIP6 OMIP simulations has been coordinated by the World Climate Research Programme (WCRP) Climate Variability and Predictability (CLIVAR) Working Group on Ocean Model Development Panel (OMDP), and ongoing research collaboration is done through the OMDP to further develop OMIP2 (Griffies et al., 2016). The same configuration is used under two different atmospheric forcing datasets as mentioned in Tsujino et al. (2020). The JRA55-do atmospheric

forcing is relatively new with major improvements, e.g., increased temporal frequency (3h) and horizontal resolution (0.5625°), compared to CORE-II forcing (6h and 1.875°). The Arctic and Antarctic sea ice concentration and drift simulations in CMIP6 OMIP2 models forced by JRA55-do are improved compared to those in OMIP1 models forced by CORE-II (Tsujino et al., 2020; Lin et al., 2021). This provides an opportunity to check the processes contributing to these improvements under changed

atmospheric forcing in the OMIP models and to compare the sea ice simulation differences in the Arctic and Antarctic.

The spatial variability of sea ice concentration and its links with the atmospheric circulation vary with season. The change in the position and strength of the cyclonic or anticyclonic circulation center over

the sea ice can affect the sea ice motion and freezing/melting processes (Rigor et al., 2002; Raphael and Hobbs, 2014; Ding et al., 2017). Strong winter wind-driven ice exports in the Eurasian coastal region occur during high North Atlantic Oscillation (NAO) index years, which can have contributed to the reduction of summer Arctic sea ice extent observed during the 1980s and 1990s (Hu et al., 2002). In the Antarctic, the decreases of sea ice concentration generally occur in regions of poleward atmospheric

flow and the increases of sea ice concentration occur in regions of equatorward flow (Renwick et al., 2012). During the seasonal sea ice advance and retreat periods, the spatial ice concentration variability is associated with different atmospheric circulation patterns and both thermal advection and dynamical forcing are important (Raphael and Hobbs, 2014). The thermodynamic and dynamic processes that

contribute to the Antarctic sea ice concentration seasonal evolution are discussed in Barthélemy et al. (2018). These authors conducted three sensitivity experiments with different atmospheric forcing fields using the NEMO-LIM3 ocean-sea ice model. They found that differences in the thermodynamic component of the forcing were mostly responsible for the differences in ice concentration simulated by the model experiments during the melting season, while during the ice expansion period, both thermodynamic and dynamic components were important. The relationships between spatially-averaged observed sea ice drift speed in the central Arctic and ice concentration, ice thickness and wind stress were investigated by Olason and Notz (2014). According to their results, on the seasonal time scales, ice drift speed changes in the central Arctic are primarily attributable to the changes in the ice concentration from June to November and changes in the ice thickness when the ice concentration is high, i.e., from December to March. The relationships between Arctic sea ice drift speed, concentration and thickness are relatively well captured by the NEMO-LIM3 model (Docquier et al., 2017) and the coupled model GFDL-ESM2G (Eyring et al., 2020), with higher drift speed associated with lower concentration and thickness. In the Antarctic, away from the coastline, the mean ice drift is significantly correlated with the wind forcing in the Pacific and Atlantic sectors, with the spatially averaged vector correlation coefficient larger than 0.7 (Kimura, 2004; Holland and Kwok, 2012).

This paper complements a companion publication (Lin et al., 2021) that documents a new Sea Ice Evaluation Tool (SITool v1.0) and applies this tool to assess the sea ice simulations in CMIP6 OMIP models. In that study, the improved Arctic and Antarctic ice concentration and drift simulations in CMIP6 OMIP2 compared to OMIP1 were highlighted from performance metrics and diagnostic spatial maps. In the present study, the thermodynamic and dynamic processes that contribute to the improved ice concentration simulation in OMIP2 compared to OMIP1 are assessed. The related surface sensible and latent heat fluxes, net shortwave and longwave radiation fluxes, as well as surface wind stress on sea ice are investigated to trace the origin of simulated sea ice differences back to the forcing datasets. Meanwhile, the sensitivity of ice drift simulation to the changes in ice concentration, ice thickness and surface wind stress are examined to help understand the factors responsible for improving the ice drift simulation. This paper is organized as follows. The CMIP6 OMIP models, observational references and atmospheric reanalysis data are described in Sect. 2. The sea ice concentration simulations and the effects of the thermodynamic and dynamic components of the atmospheric forcing are presented in Sect. 3.1. The ice drift simulation and the connections to ice concentration, ice thickness and wind stress are discussed in Sect. 3.2. Finally, in Sect. 4, conclusions and discussion are provided. Appendix A presents some extra sea ice diagnostics.

## 2 Models, observational references and atmospheric reanalysis data

Five CMIP6 OMIP models were forced by both CORE-II (OMIP1) and JRA55-do (OMIP2) reanalysis data so far, and they are marked as <model name + /C and /J>, such as CMCC-CM2-SR5/C and CMCC-CM2-SR5/J, respectively. Details on the CMIP6 OMIP models can be found in Sect. 2.2 of our previous paper (Lin et al., 2021). The sea ice components of five CMIP6-OMIP models are given in Table 1. Three of the five models (CMCC-CM2-SR5, MRI-ESM2-0, and NorESM2-LM) provide the tendencies of sea ice concentration attributed to dynamic vs. thermodynamic processes and surface wind stress on sea ice, while two of them (CMCC-CM2-SR5 and NorESM2-LM) provide surface heat fluxes (sensible and latent heat fluxes, and downward/upward shortwave and longwave radiation fluxes). The outputs from the three model groups that provided sea ice concentration tendencies are chosen to study the sea ice concentration budget and the effects of atmospheric forcing changes on the representation of surface fluxes and sea ice state. The sea ice simulations of another two models (EC-Earth3 and MIROC6) are also included in the analysis. The cross-metric analysis in Sect. 3.4 of Lin et al. (2021) shows that NorESM2-LM/J is the best performing model regarding ice concentration in both hemispheres, but the worst for ice drift. For the sake of readability and to not overload the manuscript, figures of the main text focus on this model. The other four models do not show fundamentally different behavior when the atmospheric forcing is changed and the figures from these four models are available in Appendix A.

Two sets of observational references for the sea ice concentration, thickness and ice drift are used for comparison. The two sea ice concentration products are derived from the passive microwave data by using NASA Team algorithm (NSIDC-0051, Cavalieri et al., 1996) and EUMETSAT Ocean and Sea Ice Satellite Application Facility algorithm (OSI-450, Lavergne et al., 2019), respectively. The first observed ice thickness data is derived from the measurements of ESA's Environmental Satellite (Envisat) radar altimeter (Guerreiro et al., 2017) and the second one is derived from measurements of the NASA's Ice, Cloud, and land Elevation Satellite (ICESat) Geoscience Laser Altimeter System (GLAS) (Yi and Zwally, 2009; Kurtz and Markus, 2012). The first observed ice drift product is processed by the NSIDC and enhanced by the Integrated Climate Data Center (ICDC-NSIDCv4.1, Tschudi et al., 2019) and the second ice drift data (KIMURA) is processed by Kimura et al. (2013). More information on the observational references can be found in Sect. 2.2 of Lin et al. (2021). The evaluation period is chosen according to available historical model outputs and observations and is consistent with the analysis in Lin et al. (2021). The ice concentration, concentration tendencies and their relations to surface heat fluxes and wind stress are evaluated from 1980 to 2007, while the ice drift and its links to the ice concentration, ice thickness and wind stress are assessed from 2003 to 2007.

The monthly mean surface air temperature, specific humidity, downward shortwave and longwave radiation fluxes and wind speed in CORE-II and JRA55-do reanalysis datasets from 1980 to 2007 are used to evaluate the differences between two forcing datasets.

Table 1. The details of five CMIP6-OMIP sea ice models evaluated in the study. Some information can be found on the following link: https://www.cen.uni-hamburg.de/en/icdc/data/cryosphere/cmip6-sea-ice-area.html (last access: 3 April 2023).

| Model | Sea Ice Model | Sea Ice Component | References |
|---|---|---|---|
| CMCC-CM2-SR5 | CICE4 | -Energy-conserving thermodynamics on 1 layer of snow and 4 layers of ice;<br>-Elastic-Viscous-Plastic (EVP) rheology;<br>-Ice Thickness Distribution (ITD) with 5 thickness categories;<br>-A Delta-Eddington multiple-scattering shortwave radiation treatment;<br>-Explicit level-ice melt ponds parameterization; | Hunke and Lipscomb (2008);<br>Cherchi et al. (2019); |
| EC-Earth3 | LIM3 | -Energy-conserving halo-thermodynamics with prognostic sea ice salinity on 1 layer of snow and 2 layers of ice;<br>-EVP;<br>-ITD with 5 thickness categories;<br>-Empirical albedo function, exponential attenuation of solar radiation in sea ice if no snow;<br>-Melt ponds: step reduction in albedo when Tsu = 0°C; | Rousset et al. (2015);<br>Döscher et al. (2022); |
| MIROC6 | COCO4.9 | -Energy-conserving thermodynamics on 0 layer of snow (without heat capacity) and 1 layer of ice;<br>-EVP;<br>-ITD with 5 thickness categories;<br>-Empirical albedo function;<br>-Implicit melt ponds; | Komuro et al. (2012);<br>Tatebe et al. (2019); |
| MRI-ESM2-0 | MRI.COM4.4 | -Energy-conserving thermodynamics following Mellor and Kantha (1989) on 0 layer of snow and 1 layer of ice;<br>-EVP;<br>-ITD with 5 thickness categories;<br>-The "default" CICE CCSM3 radiation scheme;<br>-Implicit melt ponds: adjust the albedo based on surface conditions; | Tsujino et al. (2010);<br>Hunke et al. (2015);<br>Yukimoto et al. (2019); |
| NorESM2-LM | CICE5.1.2 | -Mushy-layer thermodynamics with prognostic sea ice salinity on 3 layers of snow and 8 layers of ice;<br>-EVP;<br>-ITD with 5 thickness categories;<br>-A Delta-Eddington multiple-scattering shortwave radiation treatment;<br>-Explicit level-ice melt ponds parameterization. | Hunke et al. (2015);<br>Seland et al. (2020). |

**3 Results**

**3.1 Sea ice concentration**

The 1980-2007 September and February mean spatial distribution of the Arctic and Antarctic sea ice concentration from the NorESM2-LM simulations and observational reference OSI-450 compared to the observational reference NSIDC-0051 are shown in Fig. 1 and figures from the CMCC-SR5-CM2,

MRI-ESM2-0, EC-Earth3 and MIROC6 are displayed in Figs. A1 and A2. The model biases are much larger than observational differences (Fig. 1, second column). Olason and Notz (2014) suggested that, for concentrations above 80%, variations in sea ice state variables (concentration and thickness) greatly affect the ice drift speed. To study the drivers of the ice concentration and drift speed changes, we divided the regions into two parts for each month, with ice concentration larger (interior) and smaller

(exterior) than 80% in the NSIDC-0051 observational reference. The black lines in Fig. 1 exhibit September and February contours of 80% concentration in the NSIDC-0051 data. Spatial averages of the 1980-2007 September and February mean sea ice concentration biases are given in Tables 2 and 3 for the Arctic and Antarctic, respectively. The spatial averages over the interior and exterior regions are calculated with data closer than 75 km to the coast removed to reduce the spatial noise.

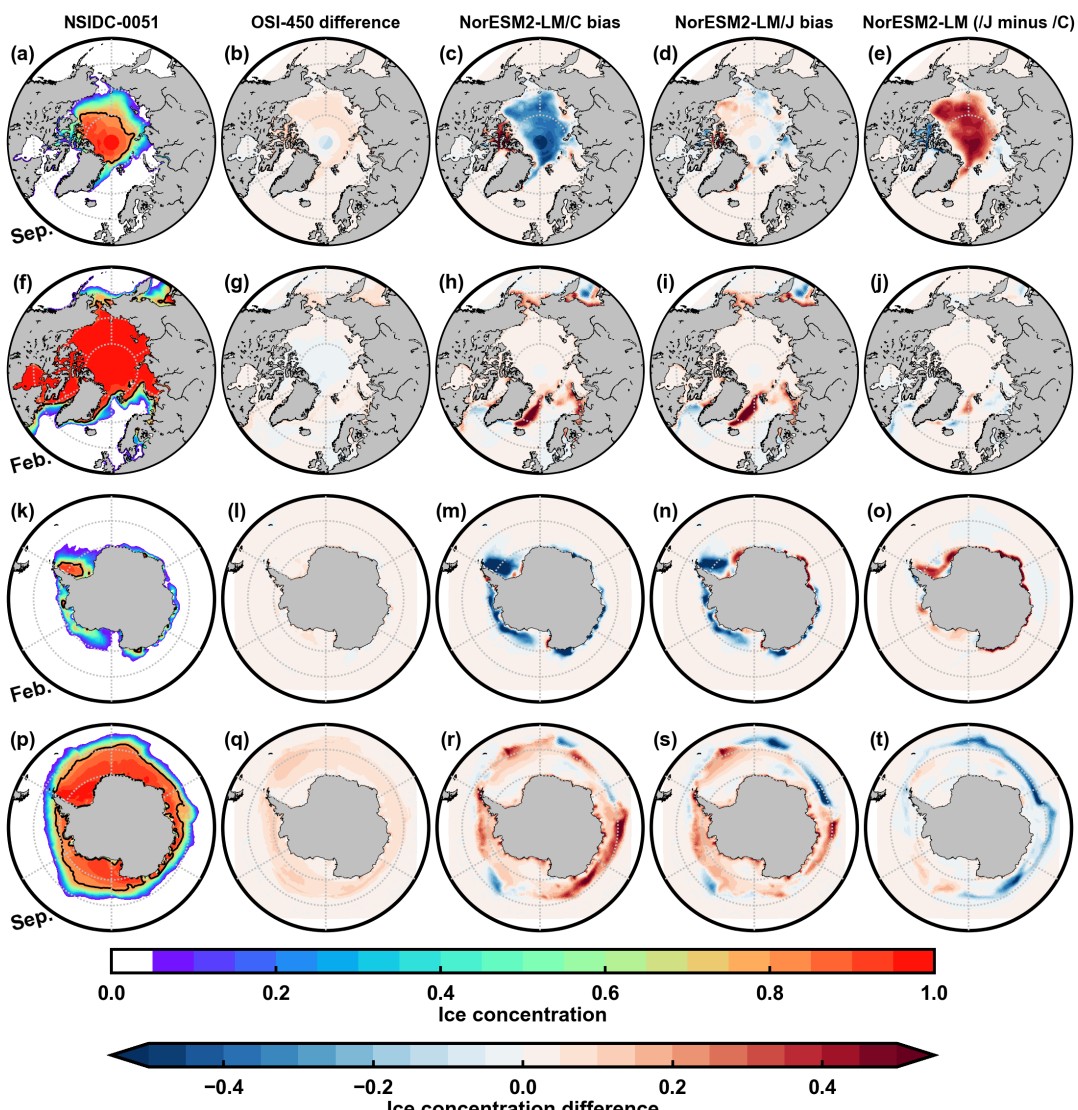

**Figure 1. 1980-2007 September and February mean Arctic (a to j) and Antarctic (k to t) sea ice concentration from the NSIDC-0051 data (first column), the differences between OSI-450 and NSIDC-0051 (second column), NorESM2-LM/C and NSIDC-0051 (third column), NorESM2-LM/J and NSIDC-0051 (fourth column), and NorESM2-LM/J and NorESM2-LM/C (fifth column). The black lines are contours of 80% concentration (a, f, k and p), which delineate**

**the interior and exterior domains to compute spatial averages in Tables 2 to 4.**

By applying the mean ice concentration difference metric developed in Lin et al. (2021), one finds that the Arctic mean ice concentration biases in OMIP1 simulations from 1980 to 2007 are reduced in OMIP2. The improvements are primarily in the boreal summer in the interior region and the Canadian Arctic Archipelago (CAA) region as shown in Figs. 1, A1 and A2. In September, the ice concentration is primarily underestimated in the interior region and overestimated in the CAA region in OMIP1 simulations and those biases are reduced in OMIP2 runs. The spatial mean ice concentration biases in the interior region are reduced from -0.31 to 0.02 in NorESM2-LM, from -0.52 to -0.16 in CMCC-SR5-CM2 and from -0.07 to -0.01 in MIROC6 in by changing the atmospheric forcing from CORE-II to JRA55-do (Table. 2). The reduced negative ice concentration bias in MRI-ESM2-0 and EC-Earth3 are over the eastern part of the central Arctic Ocean. The spatial mean ice concentration bias in the interior region in MRI-ESM2-0/C (-0.03) and EC-Earth3/C (-0.04) are not weakened in MRI-ESM2-0/J (0.06) and EC-Earth3/J (0.04). The spatial mean ice concentration biases in the CAA are reduced from 0.26 to 0.07 in NorESM2-LM, from 0.1 to -0.03 in CMCC-SR5-CM2, from 0.13 to 0.01 in MRI-ESM2-0, from 0.32 to 0.18 in EC-Earth3 and from 0.29 to 0.27 in MIROC6 under changed forcing from CORE-II to JRA55-do. In February, the ice concentration biases in exterior regions in five OMIP1 simulations are also present in OMIP2 runs, with minor reduction.

The Antarctic mean ice concentration biases in OMIP1 simulations from 1980 to 2007 are also diminished. The improvements are mainly over the coastal regions of the western Weddell Sea and the Amundsen Sea in the austral summer and over exterior regions from 70° to 180°E in winter from OMIP1 to OMIP2 as shown in Figs. 1, A1 and A2. In February, the spatial mean ice concentration biases in the interior region from 52° to 60°W are reduced from -0.71 to -0.41 in NorESM2-LM, from -0.68 to -0.51 in CMCC-SR5-CM2, from -0.84 to -0.76 in MRI-ESM2-0, from -0.66 to -0.54 in EC-Earth3 and from -0.87 to -0.60 in MIROC6 by changing the atmospheric forcing from CORE-II to JRA55-do (Table. 3). Positive ice concentration biases are shown in the central Weddell Sea (CWS) in CMCC-SR5-CM2/C, MRI-ESM2-0/C, EC-Earth3/C but not in NorESM2-LM/C and MIROC6/C. The spatial mean ice concentration biases over the CWS are reduced from 0.36 to 0.22 in CMCC-SR5-CM2, from 0.08 to 0.01 in MRI-ESM2-0 and from 0.02 to -0.0003 in EC-Earth3 under changed forcing from CORE-II to JRA55-do. The NorESM2-LM exhibits a larger positive bias on the East Antarctic coast when forced by JRA55-do as compared with CORE-II. In September, the spatial mean ice concentration biases in exterior regions from 70° to 180°E are reduced from 0.24 to 0.09 in NorESM2-LM, from 0.19 to 0.10 in CMCC-SR5-CM2, from 0.33 to 0.20 in MRI-ESM2-0, from 0.30 to 0.21 in EC-Earth3 and from 0.20 to 0.11 in MIROC6 under changed forcing from CORE-II to JRA55-do.

235

**Table 2.** Spatial averages of the 1980-2007 September mean Arctic sea ice concentration (SIC) biases (vs. NSIDC-0051, Figs. 1, A1, A2), as well as the ice concentration tendencies through thermodynamic and dynamic processes (Figs. 2, 3, A3 and A4), surface heat fluxes and surface stress on sea ice (Figs. 4, A5) over March-August. The results derived from five model groups under OMIP1 [/C] and OMIP2 [/J] runs are listed. The spatial averages over the interior region in the Arctic and the Canadian Arctic Archipelago (CAA) in summer are given. The improvements on ice concentration simulations and the contributions from thermodynamic process and the surface heat flux on sea ice are marked in blue and green for the interior region and the CAA region, respectively.

| Arctic | | | | | | | |
|---|---|---|---|---|---|---|---|
| Variables | Periods | Regions | NorESM2-LM/C [/J] | CMCC-SR5-CM2/C [/J] | MRI-ESM2-0/C [/J] | EC-Earth3/C [/J] | MIROC6/C [/J] |
| SIC bias | Sep. | Interior | -0.31[0.02] | -0.52[-0.16] | -0.03[0.06] | -0.04[0.04] | -0.07[-0.01] |
| | | CAA | 0.26[0.07] | 0.10[-0.03] | 0.13[0.01] | 0.32[0.18] | 0.29[0.27] |
| siconc tendency thermo. ($10^{-3}$ day$^{-1}$) | Mar.-Aug. | Interior | -2.2[0.6] | -3.2[-0.9] | 0.4[1.5] | / | |
| | | CAA | -0.05[-1.05] | -0.7[-1.4] | 1.0[0.22] | | |
| siconc tendency dyn. ($10^{-3}$ day$^{-1}$) | Mar.-Aug. | Interior | -1.6[-1.9] | -1.4[-1.6] | -2.0[-2.2] | | |
| | | CAA | -1.6[-1.7] | -1.6[-1.63] | -2.1[-2.19] | | |
| surface heat flux on sea ice (downward positive, W m$^{-2}$) | Mar.-Aug. | Interior | 27.4[14] | 32.9[19.1] | / | | |
| | | CAA | 24.5[40.2] | 28.7[46.2] | / | | |
| surface stress on sea ice ($10^{-3}$ N m$^{-2}$) | Mar.-Aug. | Interior | 11.9[12.6] | 11.5[12.3] | 12.2[12.6] | | |
| | | CAA | 16.6[16.8] | 17.4[18.9] | 19.8[20.3] | | |

### 3.1.1 Effects of thermodynamic vs. dynamic processes to ice concentration tendencies

To understand the differences in the simulated sea ice concentration noted in Figs. 1, A1 and A2, we analyze the thermodynamic and dynamic processes contributing to the concentration tendencies during the melt and growth seasons under different atmospheric forcings (Figs. 2, 3, A3, A4). The idea is close to the sea ice concentration budget proposed in Holland and Kwok (2012) and applied in Uotila et al. (2014), Lecomte et al. (2016) and Barthélemy et al. (2018). The contributing thermodynamic processes to the concentration tendencies are freezing or melting, whereas the relevant dynamic processes are ice advection, divergence/convergence and mechanical redistribution (rafting/ridging). The tendencies of ice concentration due to dynamic and thermodynamic processes are available as standard SIMIP diagnostics in the three models (Notz et al., 2016). Spatial averages of the Arctic and Antarctic ice concentration tendencies due to thermodynamic and dynamic processes are listed in Tables 2 and 3, respectively.

**Table 3.** Spatial averages of the 1980-2007 February and September mean Antarctic SIC biases (vs. NSIDC-0051, Figs. 1, A1, A2), as well as the ice concentration tendencies through thermodynamic and dynamic processes (Figs. 2, 3, A3 and A4), surface heat fluxes and surface stress on sea ice (Figs. 4, A5) over October-January and March-August. The results derived from five model groups under OMIP1 [/C] and OMIP2 [/J] runs are listed. The spatial averages over the interior region in the Antarctic (52° to 60°W) and central Weddell Sea (CWS) in summer and over the exterior region in the Antarctic (70° to 180°E) in winter are given. The improvements on ice concentration simulations and the contributions from thermodynamic process and the surface heat flux on sea ice are marked in blue and green for the interior region and the CWS region, respectively. The improvements on ice concentration simulations of the exterior region and the contributions from dynamic process and the surface stress on sea ice are marked in orange.

| Antarctic | | | | | | | |
|---|---|---|---|---|---|---|---|
| Variables | Periods | Regions | NorESM2-LM/C [/J] | CMCC-SR5-CM2/C [/J] | MRI-ESM2-0/C [/J] | EC-Earth3/C [/J] | MIROC6/C [/J] |
| SIC bias | Feb. | Interior | -0.71[-0.41] | -0.68[-0.51] | -0.84[-0.76] | -0.66[-0.54] | -0.87[-0.60] |
| | | CWS | 0.005[0.003] | 0.36[0.22] | 0.08[0.01] | 0.023[-0.0003] | -0.0003[-0.0003] |
| | Sep. | Exterior | 0.24[0.09] | 0.19[0.1] | 0.33[0.2] | 0.3[0.21] | 0.2[0.11] |
| siconc tendency thermo.($10^{-3}$ day$^{-1}$) | Oct.-Jan. | Interior | -4.5[-1.9] | -5.2[-2.9] | -6.1[-3.9] | / | |
| | | CWS | -6.1[-6.12] | -4.4[-7.2] | -3.0[-5.2] | | |
| | Mar.-Aug. | Exterior | 1.1[2.7] | 0.5[1.4] | 3.1[4.2] | | |
| siconc tendency dyn. ($10^{-3}$ day$^{-1}$) | Oct.-Jan. | Interior | -1.2[-2.1] | -0.3[-1.2] | -1.3[-3] | | |
| | | CWS | -1.3[-1.33] | -1.0[-0.2] | -3.7[-2.6] | | |
| | Mar.-Aug. | Exterior | 2.9[0.5] | 3.1[1.7] | 1.2[-0.6] | | |
| surface heat flux on sea ice (downward positive, W m$^{-2}$) | Oct.-Jan. | Interior | 32.4[23.1] | 31.6[27.8] | / | | |
| | | CWS | 7.1[26.3] | 4.6[34.1] | / | | |
| | Mar.-Aug. | Exterior | -19.5[-16.3] | -14.1[-6.5] | / | | |
| surface stress on sea ice ($10^{-3}$ N m$^{-2}$) | Oct.-Jan. | Interior | 24.3[24.9] | 24.6[25] | 31.3[26.5] | | |
| | | CWS | 26.8[25.3] | 35.1[29.1] | 41.9[39.6] | | |
| | Mar.-Aug. | Exterior | 29.1[22.7] | 20.4[18.5] | 75.5[68.8] | | |

Compared to OMIP1 runs, changes in thermodynamic processes in the Arctic Ocean and the CAA region contribute to the ice concentration changes in OMIP2 runs during March to August (Figs. 2 and

A3). The differences between OMIP1 and OMIP2 simulations on the contribution from dynamic processes are small. As shown in Table 2, the spatial mean ice concentration tendencies due to thermodynamic processes (unit: $10^{-3}$ day$^{-1}$) from OMIP1 to OMIP2 simulations are increased in the interior region (from -2.2 to 0.6 in NorESM2-LM and -3.2 to -0.9 in CMCC-SR5-CM2) and decreased in the CAA (from -0.05 to -1.05 in NorESM2-LM, -0.7 to -1.4 CMCC-SR5-CM2 and 1.0 to 0.22 in MRI-ESM2-0). This is consistent with the reduced September Arctic ice concentration biases in OMIP2 runs (Figs. 1c to e, A1a to f). That is, by changing the atmospheric forcing from CORE-II to JRA55-do, the simulations of Arctic summer ice concentration in the Arctic Ocean and the CAA region are improved owing to a better representation of the thermodynamic processes. The thermodynamic processes in MRI-ESM2-0/C contribute to the increase of ice concentration in the Beaufort Gyre region (Fig. A3e) but not the decrease in the other two models (Figs. 2b and A3b) during March to August. This explains why the large negative ice concentration biases in the Beaufort Gyre region in the other two models are not present in MRI-ESM2-0/C (Figs. 1c to e, A1a to f).

The major Arctic winter ice concentration biases are located in the exterior regions in OMIP1 simulations, with a minor reduction in OMIP2 runs (Figs. 1h to j, A1g to l). The winter ice concentration simulation in exterior regions is complicated because both dynamic and thermodynamic processes are important. The contributions from thermodynamic and dynamic processes are anti-correlated in these regions, with the dynamic processes increasing ice concentration through the expansion of sea ice and the thermodynamic processes contributing to ice melt. During October to January, the increased Arctic ice concentration is dominated by dynamic processes in exterior regions in OMIP1 simulations (Fig. 2 and A3). Compared to OMIP1 runs, these dynamic processes in OMIP2 runs contribute to the decreased ice concentration in exterior regions on the east of Greenland, while thermodynamic processes contribute to the increased ice concentration. This contributes to minor winter ice concentration differences between OMIP1 and OMIP2 simulations.

During October to January, the thermodynamic processes contribute to the decreased ice concentration in the Southern Ocean, except in some coastal regions, and the dynamic processes contribute to the decreased ice concentration in the inner region in the three OMIP1 runs (Figs. 3 and A4). Compared to OMIP1 runs, the thermodynamic processes dominate the increased ice concentration in the coastal regions of the western Weddell Sea and Amundsen Sea in the three OMIP2 simulations, as well as the decreased ice concentration in the CWS in CMCC-SR5-CM2/J and MRI-ESM2-0/J. As shown in Table 3, the spatial mean ice concentration tendency due to thermodynamic processes (unit: $10^{-3}$ day$^{-1}$) from OMIP1 to OMIP2 simulations is increased in the interior region from 52° to 60°W  (from -4.5 to -1.9 in NorESM2-LM, -5.2 to -2.9 in CMCC-SR5-CM2 and -6.1 to -3.9 in MRI-ESM2-0) and decreased in the CWS (from -4.4 to -7.2 in CMCC-SR5-CM2 and -3.0 to -5.2 in MRI-ESM2-0). This is consistent with

the reduced February Antarctic ice concentration biases in OMIP2 runs (Figs. 1m to o, A1m to r). The positive bias in the coastal region of the East Antarctic sector in the NorESM2-LM/J simulation (Fig. 1n) is related to thermodynamic processes (Fig. 3e). By changing the atmospheric forcing from CORE-II to JRA55-do, the simulations of Antarctic summer ice concentration in the coastal regions of the western Weddell Sea and the Amundsen Sea, as well as the CWS region, are improved owing to the thermodynamic processes.

During March to August, the thermodynamic processes contribute to the increased Antarctic ice concentration, except for some exterior regions, and the dynamic processes contribute to the increased ice concentration primarily in the exterior region (Figs. 3 and A4). Compared to OMIP1 runs, the dynamic processes dominate the decreased ice concentration in exterior regions in the three OMIP2 simulations. As shown in Table 3, the spatial mean ice concentration tendencies related to dynamic processes (unit: $10^{-3}$ $day^{-1}$) from OMIP1 to OMIP2 simulations are decreased in the exterior region from 70° to 180°E (from 2.9 to 0.5 in NorESM2-LM, 3.1 to 1.7 in CMCC-SR5-CM2 and 1.2 to -0.6 in MRI-ESM2-0). This is consistent with the reduced September Antarctic ice concentration biases from 70° to 180°E in OMIP2 runs (Figs. 1r to t, A1s to x). The simulations of the Antarctic winter ice concentration in the exterior region from 70° to 180°E are improved due to the dynamic processes when forced by JRA55-do as compared with CORE-II.

In general, by changing the atmospheric forcing from CORE-II to JRA55-do, the improvements in the simulation of summer Arctic and Antarctic sea ice concentration within the pack are driven by differences in the thermodynamic tendency terms, while the improvements in Antarctic winter concentration simulation in the exterior region from 70° to 180°E are dominated by differences in dynamic tendency terms. For other cases (winter Arctic ice concentration in the exterior region, ice concentration in coastal regions), improvements are not as clear.

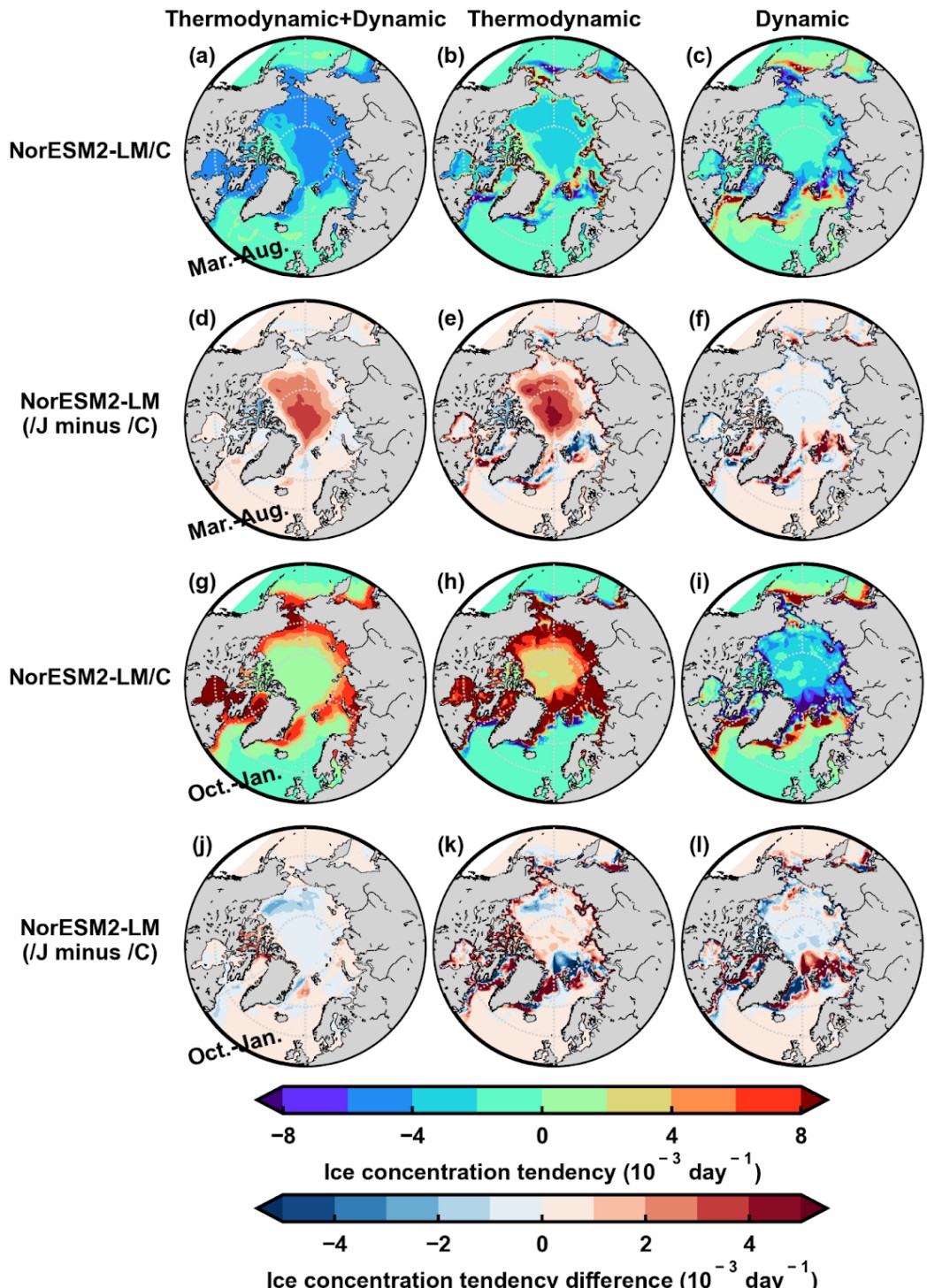

**Figure 2. 1980-2007 March–August (a to f) and October–January (g to l) mean Arctic sea ice concentration tendencies**

**in NorESM2-LM/C (a to c, g to i) and the differences between NorESM2-LM/J and NorESM2-LM/C (d to f, j to l).**

**The ice concentration tendencies due to thermodynamic and dynamic processes in total are in the first column, and**

**individual contributions are in the second and third columns.**

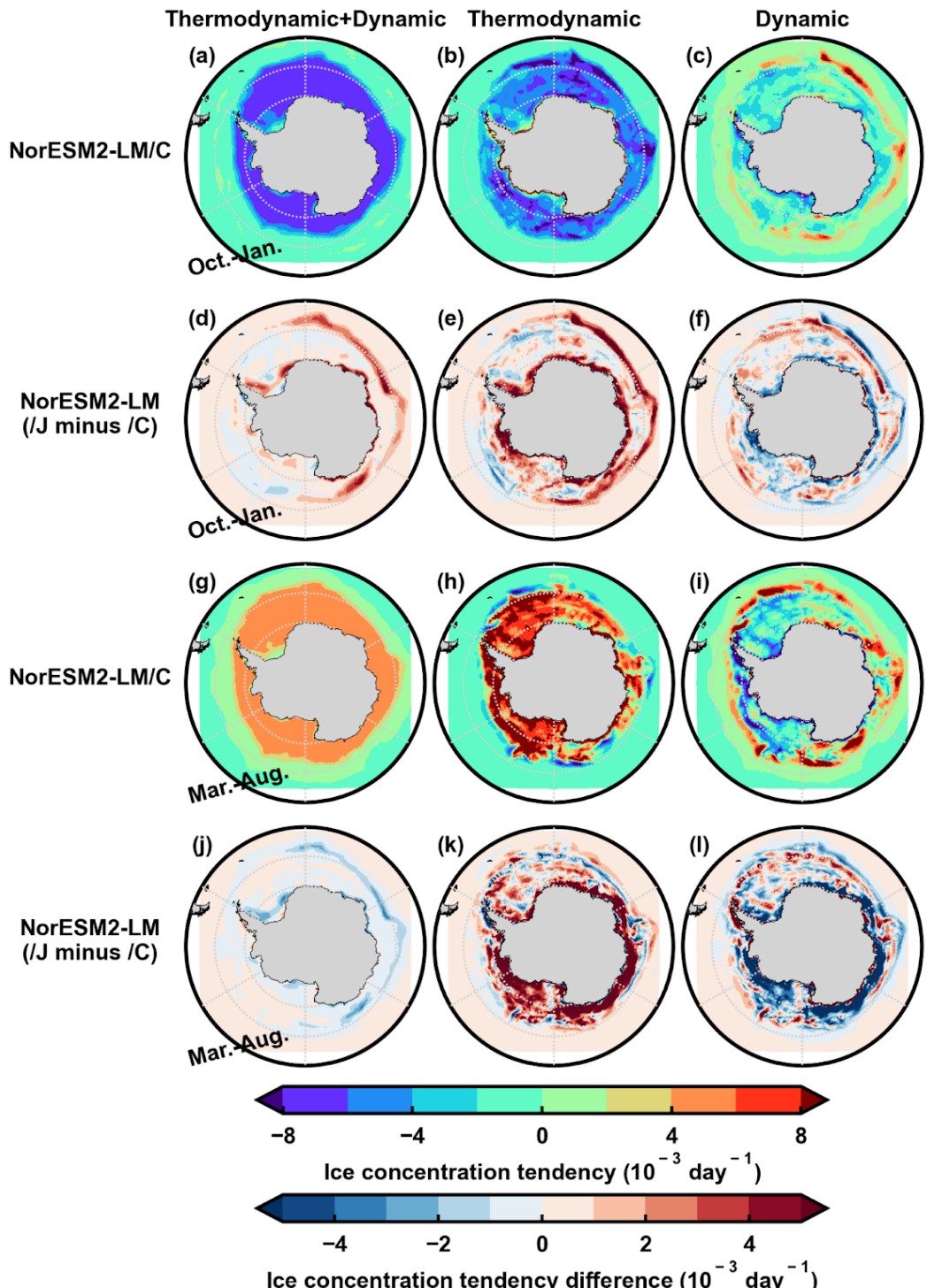

**Figure 3. Same as Fig. 2 but for the 1980-2007 October-January (a to f) and March-August (g to l) mean Antarctic**
**sea ice concentration tendencies.**

## 3.1.2 Surface heat and momentum flux

To trace the origin of the differences in thermodynamic and dynamic tendency terms noted in the previous section, the surface heat and momentum fluxes available from the standard OMIP1 and 345 OMIP2 model outputs are compared. The sign convention for flux in this study is that a downward flux towards the surface is positive. The net surface heat flux is downward (positive) in the Arctic during

March to August and in the Antarctic during October to January in OMIP1 runs (Figs. 4 and A5). Compared to OMIP1 simulations, the net surface heat fluxes in OMIP2 simulations are smaller in the central Arctic Ocean and over the coastal regions of the western Weddell Sea, and larger in the CAA and CWS regions. As shown in Tables 2 and 3, the spatial mean net surface heat flux from OMIP1 to OMIP2 simulations is decreased in the Arctic interior region (from 27.4 to 14 W m$^{-2}$ in NorESM2-LM and 32.9 to 19.1 W m$^{-2}$ in CMCC-SR5-CM2) and in the Antarctic interior region from 52° to 60°W (from 32.4 to 23.1 W m$^{-2}$ in NorESM2-LM and 31.6 to 27.8 W m$^{-2}$ in CMCC-SR5-CM2), and increased in the CAA region (from 24.5 to 40.2 W m$^{-2}$ in NorESM2-LM and 28.7 to 46.2 W m$^{-2}$ in CMCC-SR5-CM2) and the CWS region (4.6 to 34.1 W m$^{-2}$ in CMCC-SR5-CM2). The net surface heat flux changes in OMIP2 simulations contribute to the improved ice concentration simulation in those regions (Figs. 1 and A1). The simulated surface fluxes on sea ice are compared to ERA5 reanalysis data, which is the fifth generation ECMWF reanalysis for the global atmosphere, land surface and ocean waves (Hersbach et al., 2018). The net surface heat flux in OMIP2 simulations in these regions is close to the net surface heat flux on sea ice derived from ERA5 12-hourly data (not shown).

To study which part dominates the surface heat flux changes from OMIP1 to OMIP2 simulations, the surface sensible and latent heat fluxes and the net shortwave and longwave radiation fluxes are computed (Figs. 5 and A6). Compared to OMIP1 simulations, the net shortwave radiation flux and latent heat flux in OMIP2 are smaller in the central Arctic Ocean and the coastal region of the western Weddell Sea, the net shortwave radiation flux is larger in the CAA and CWS regions and the sensible heat flux is larger in the CAA region. As shown in Table 4, the decreased net shortwave radiation flux in OMIP2 simulations (-13.4 W m$^{-2}$ in NorESM2-LM and -12.7 W m$^{-2}$ in CMCC-SR5-CM2) dominates the net surface heat flux changes in the central Arctic Ocean. The decreased latent heat flux in OMIP2 simulations (-12.6 W m$^{-2}$ in NorESM2-LM and -13.1 W m$^{-2}$ in CMCC-SR5-CM2) dominates the net surface heat flux changes in the coastal region of the western Weddell Sea. In the CAA region, the increased sensible heat flux and net shortwave radiation flux in OMIP2 simulations (16.1 and 10.0 W m$^{-2}$ in NorESM2-LM, 14.9 and 13.6 W m$^{-2}$ in CMCC-SR5-CM2) dictate the net surface heat flux changes. In the CWS region, the increased net shortwave radiation flux (19.9 W m$^{-2}$ in NorESM2-LM and 39.2 W m$^{-2}$ in CMCC-SR5-CM2) is the major contributor to the net surface heat flux changes.

The changes in the shortwave radiation flux are crucial for the summer ice concentration changes in the OMIP2 simulations in the Arctic interior region and the CAA and CWS regions. The downward and upward shortwave radiation fluxes in NorESM2-LM and CMCC-SR5-CM2 (Fig. A7) as well as spatial averages (Table 4) are displayed. The decreased downward shortwave radiation flux in OMIP2 simulations (-9.6 W m$^{-2}$ in NorESM2-LM and -9.9 W m$^{-2}$ in CMCC-SR5-CM2) is mostly responsible for the net shortwave radiation flux changes in the central Arctic Ocean. The increased net shortwave

radiation fluxes in the CAA and CWS regions are related to the increased downward and decreased upward shortwave radiation flux in OMIP2 simulations.

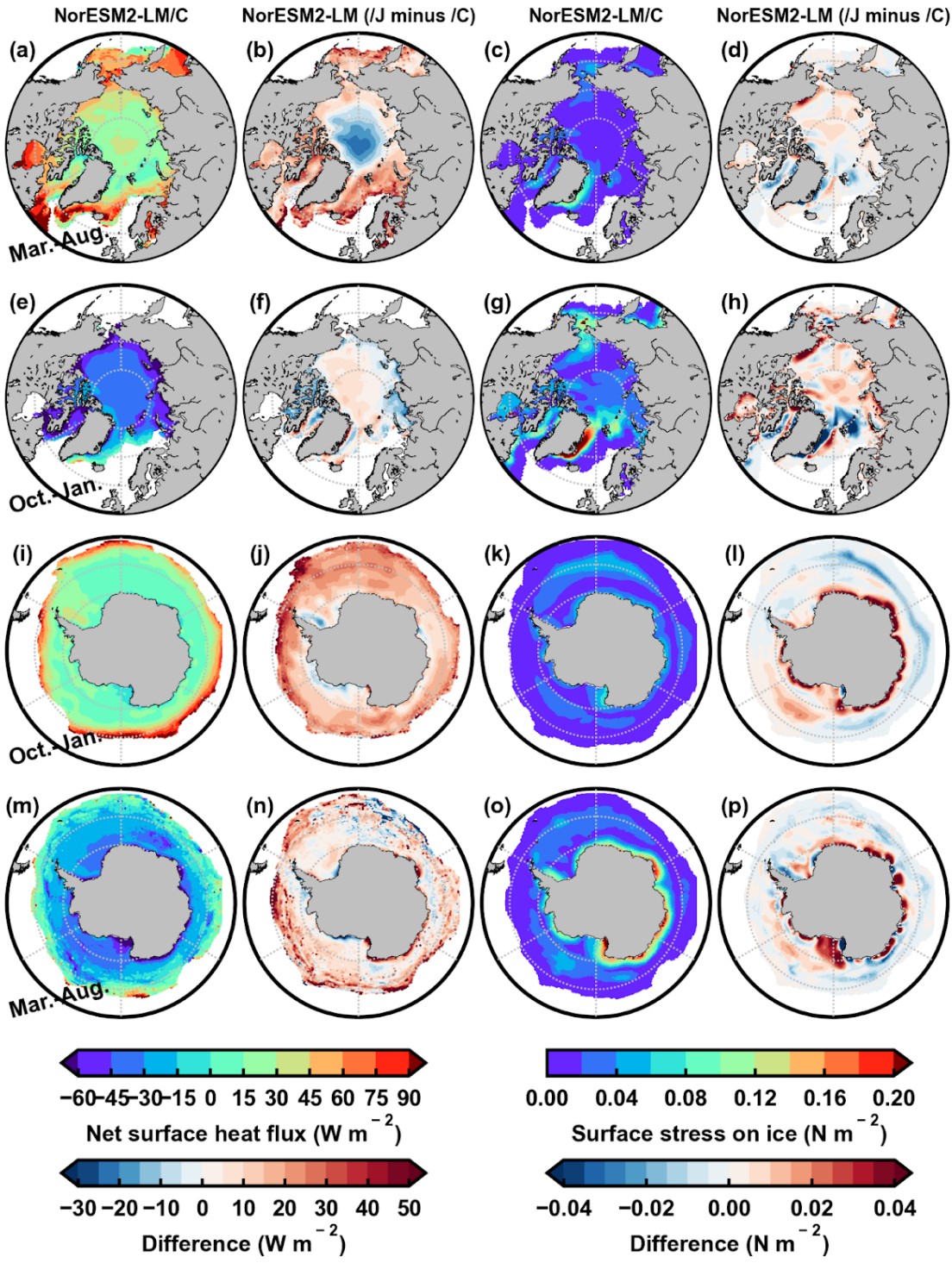

**Figure 4. 1980-2007 March-August and October-January mean Arctic (a to h) and Antarctic (i to p) net surface heat flux (first two columns) and surface stress (last two columns) on sea ice. The positive values indicate a surface flux downward. The first and third columns correspond to NorESM2-LM/C, and the second and fourth columns are differences between NorESM2-LM/J and NorESM2-LM/C.**

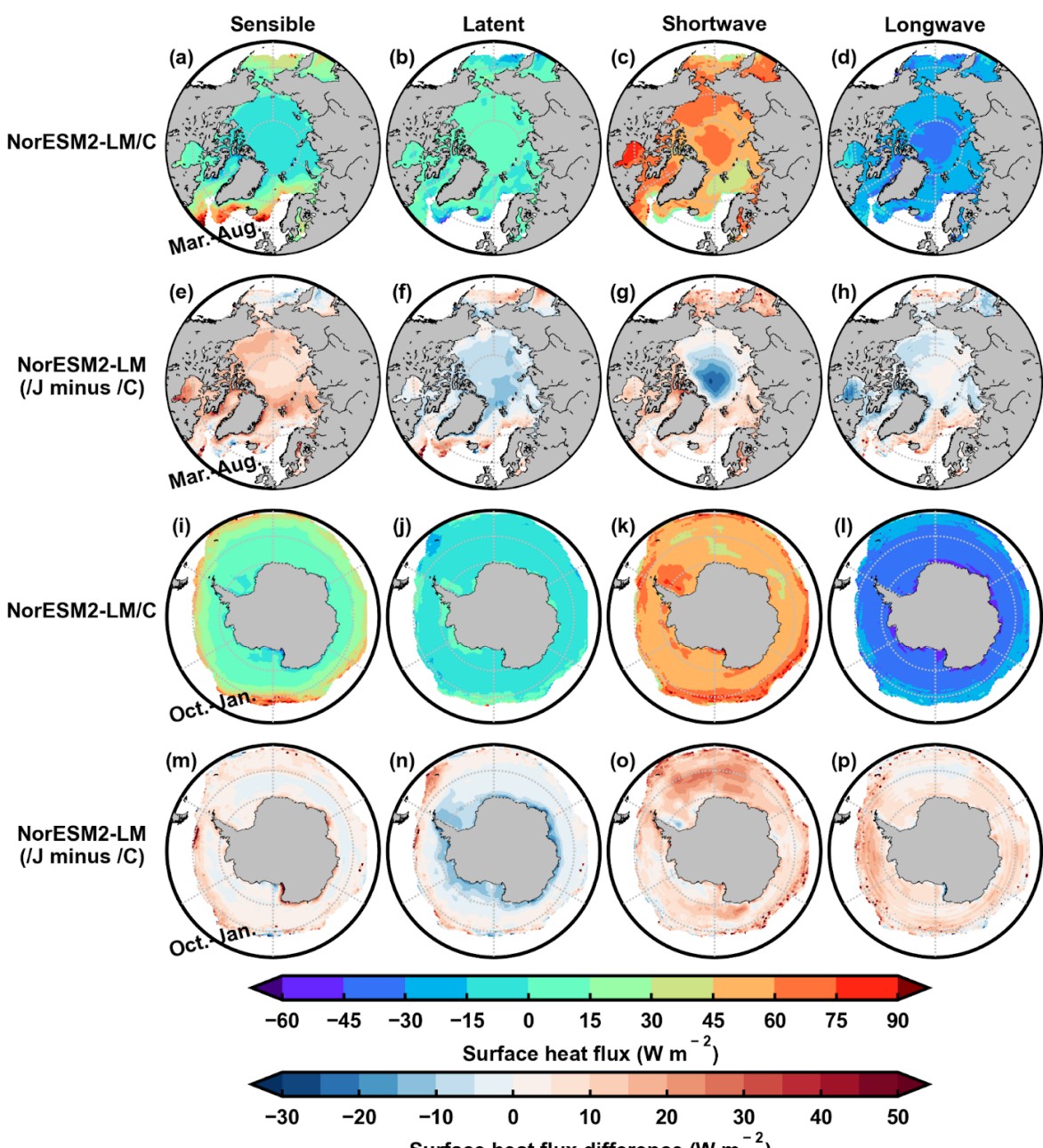

**Figure 5. 1980-2007 March-August mean Arctic (a to h) and October-January mean Antarctic (i to p) surface sensible (first column) and latent heat fluxes (second column), net shortwave (third column) and longwave radiation fluxes (fourth column). The positive values indicate a surface flux downward. The first and third rows correspond to NorESM2-LM/C, and the second and fourth rows are differences between NorESM2-LM/J and NorESM2-LM/C.**

**Table 4.** Spatial averages of the 1980-2007 mean Arctic (March-August) and Antarctic (October-January) net surface heat flux (Figs. 4 and A5), sensible and latent heat fluxes, net shortwave and longwave radiation fluxes (Figs. 5 and A6), downward and upward shortwave fluxes (Figs. A7), as well as downward and upward longwave fluxes over the interior region in the Arctic and Antarctic (52° to 60°W), the CAA and CWS regions. The results derived from two model groups under OMIP1 [/C] and OMIP2 [/J] runs are listed. The contributions from the surface heat flux to the improved ice concentration simulations are marked in blue and green for the interior region, and the CAA and CWS regions, respectively.

| Periods & regions | Variables (W m$^{-2}$) downward positive | Arctic | | | | Antarctic | | | |
|---|---|---|---|---|---|---|---|---|---|
| | | NorESM2-LM /C [/J] | | CMCC-SR5-CM2 /C [/J] | | NorESM2-LM /C [/J] | | CMCC-SR5-CM2 /C [/J] | |
| Interior | net surface heat flux | 27.4[14.0] | | 32.9[19.1] | | 32.4[23.1] | | 31.6[27.8] | |
| | sensible | -7.0[3.9] | | -6.9[3.4] | | 2.7[4.9] | | 5.4[6.5] | |
| | latent | 4.8[-4.4] | | 4.6[-5.0] | | 0.8[-11.8] | | 1.7[-11.4] | |
| | net shortwave | 60.8[47.4] | | 66.4[53.7] | | 71.5[67.3] | | 66.5[69.8] | |
| | downward shortwave / upward shortwave | 177.7 [168.1] | -116.9 [-120.7] | 178.8 [168.9] | -112.5 [-115.3] | 238.2 [238.7] | -166.6 [-171.3] | 237.6 [238.2] | -171.1 [-168.4] |
| | net longwave | -31.2[-32.9] | | -31.2[-33.0] | | -42.5[-37.3] | | -41.9[-37] | |
| | downward longwave / upward longwave | 242.3 [238.4] | -273.4 [-271.2] | 242.3 [238.6] | -273.5 [-271.6] | 253.0 [251.0] | -295.5 [-288.3] | 252.4 [250.4] | -294.3 [-287.4] |
| CAA/CWS | net surface heat flux | 24.5[40.2] | | 28.7[46.2] | | 7.1[26.3] | | 4.6[34.1] | |
| | sensible | -6.6[9.5] | | -5.6[9.3] | | 8.5[8.4] | | 13.1[9.1] | |
| | latent | 5.2[-1.7] | | 5.2[-2.2] | | -10.5[-12.9] | | -8.2[-13.3] | |
| | net shortwave | 53.6[63.6] | | 56.5[70.1] | | 46.2[66.1] | | 34.3[73.5] | |
| | downward shortwave / upward shortwave | 172.7 [178.7] | -119.1 [-115.1] | 173.3 [177.8] | -116.8 [-107.7] | 215.0 [224.8] | -168.8 [-158.6] | 213.5 [223.7] | -179.2 [-150.2] |
| | net longwave | -27.7[-31.1] | | -27.4[-30.9] | | -37.1[-35.3] | | -34.5[-35.2] | |
| | downward longwave / upward longwave | 246.1 [240.3] | -273.8 [-271.4] | 246.2 [240.6] | -273.7 [-271.6] | 260.1 [262.8] | -297.2 [-298.2] | 260.5 [262.8] | -295.0 [-297.9] |

Compared to other regions, the surface stress on ice along the east coasts of Greenland, Svalbard and Baffin Island, near the Bering Strait from 60 to 70°N, in the Antarctic coastal regions and inside the subpolar gyres is larger in OMIP1 simulations (third column in Fig. 4 and third and fifth columns in Fig. A5). This contributes to the smaller ice concentration due to the dynamic processes in those regions

(Figs. 2, 3, A3 and A4). The large winter concentration biases in both hemispheres are located in exterior regions. In wintertime, the reduced Antarctic ice concentration biases in the exterior region from 70° to 180°E in OMIP2 simulations are dominated by the dynamic processes, as discussed in the previous section (Figs. 3 and A4). Compared to OMIP1 simulations, the surface wind stress on Antarctic sea ice in OMIP2 simulations is weaker in the inner part of the exterior region from 70° to 180°E (Figs. 4o and p, A5u to x). As shown in Table 3, the spatial mean surface wind stress on sea ice is decreased from OMIP1 to OMIP2 simulations in the exterior region from 70° to 180°E (from 29.1 to 22.7 N m$^{-2}$ in NorESM2-LM, 20.4 to 18.5 N m$^{-2}$ in CMCC-SR5-CM2 and 75.5 to 68.8 N m$^{-2}$ in MRI-ESM2-0). The decreased surface wind stress in OMIP2 simulations in the inner part of the exterior region weakens the ice motion and reduces the ice concentration in the exterior region from 70° to 180°E. The surface wind stress in OMIP2 simulations in the inner part of the exterior region is close to the surface stress on sea ice derived from ERA5 hourly data (not shown).

The improvement in the winter Arctic ice concentration in the exterior region is not as clear. Compared to OMIP1 simulations, the surface wind stress in OMIP2 simulations is smaller along the east coasts of Greenland, Svalbard and Baffin Island (Figs. 4g and h, A5i to l). This is consistent with the decrease of ice concentration in the exterior region in OMIP2 simulations due to the dynamic processes away from the east coast (Figs. 2l, A3u and A3x). However, the thermodynamic processes in OMIP2 simulations contribute to the increase in ice concentration, which is close to the decrease due to the dynamic processes in these regions (Figs. 2k, A3t and A3w). The different contributions of the thermodynamic processes to the winter ice concentration tendency in the exterior region between OMIP1 and OMIP2 simulations are primarily related to the dynamic processes, while the surface heat flux difference on the sea ice is small.

To identify how the differences in the atmospheric forcings are transferred to the model results, the 1980-2007 mean surface air temperature, specific humidity, downward shortwave and longwave radiation fluxes during melting months, and wind speed during freezing months in CORE-II and JRA55-do are shown in Fig. 6. The selection to show these variables during melting/freezing months is because in general the ice concentration simulations are improved from OMIP1 to OMIP2 in summer due to surface heat flux changes and in winter due to wind stress changes. Compared to CORE-II, the downward shortwave radiation flux and specific humidity in the central Arctic Ocean (Figs. 6g and h) and specific humidity the coastal region of the western Weddell Sea (Fig. 6q) in JRA5-do are smaller, the downward shortwave radiation flux in the CAA and CWS regions (Figs. 6h and r) and the air temperature in the CAA region are larger (Fig. 6f), and the surface wind speed on Antarctic sea ice in the inner part of the exterior region from 70° to 180°E is weaker (Fig. 6t). These differences in the atmospheric forcing are transferred to the surface fluxes and contribute to the improved ice

concentration simulation in those regions. Compared to OMIP1 simulations, the downward shortwave radiation flux and latent heat flux in the central Arctic Ocean and latent heat flux in the coastal region of the western Weddell Sea in OMIP2 are smaller, the downward shortwave radiation flux in the CAA and CWS regions and the sensible heat flux in the CAA region are larger (Figs. 5, A6, A7, Table 4), and the surface wind stress on Antarctic sea ice in the inner part of the exterior region from 70° to 180°E is weaker (Figs. 4, A5, Table 3).

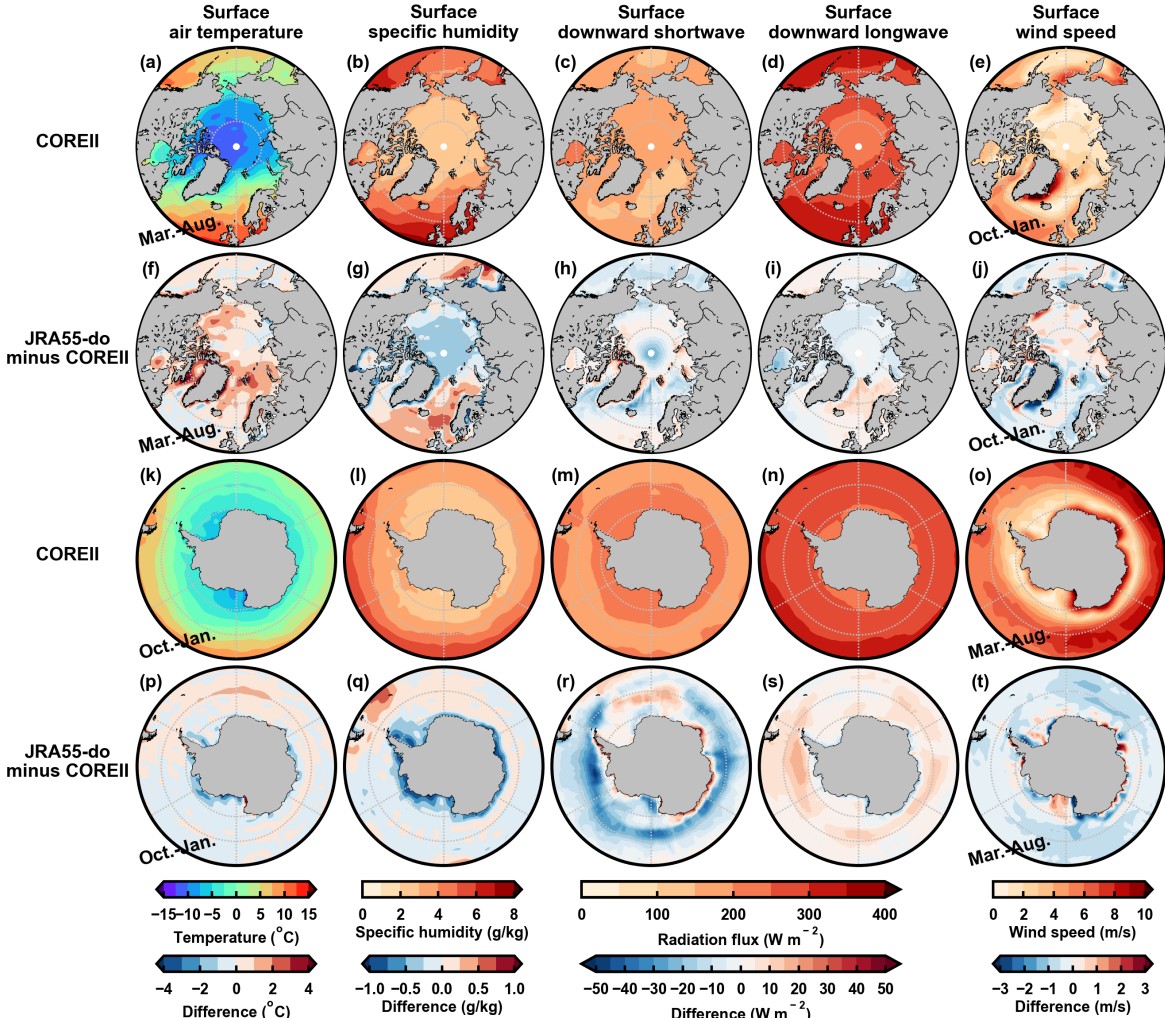

**Figure 6. 1980-2007 March-August mean Arctic and October-January mean Antarctic surface air temperature (first column) and specific humidity (second column), downward shortwave (third column) and longwave radiation fluxes (fourth column), as well as October-January mean Arctic and March-August mean Antarctic surface wind speed (fifth column). The first and third rows correspond to CORE-II, and the second and fourth rows are differences between JRA55-do and CORE-II.**

## 3.2 Sea ice drift

### 3.2.1 Ice drift magnitude and its links with ice concentration, ice thickness and wind stress

The Arctic and Antarctic ice drift magnitude and direction simulations are improved from OMIP1 to OMIP2 (Lin et al., 2021). To understand the factors responsible for this feature, the sensitivity of the ice

drift magnitude simulation to the changes in ice concentration, ice thickness and surface wind stress is investigated. The mean kinetic energy (MKE) is calculated to measure the ice drift magnitude,

$$MKE = \frac{1}{2}(u^2 + v^2),$$  (1)

where u and v are zonal and meridional components of ice drift, respectively. The simulated monthly mean and spatially averaged values of the ice MKE and their links with the ice concentration, ice thickness and surface wind stress are examined for NorESM2-LM (Arctic in Fig. 7 and Antarctic in Fig.

8) and CMCC-CM2-SR5 and MRI-ESM2-0 (Arctic in Fig. A8 and Antarctic in Fig. A9). Spatial averages are computed for the interior and exterior regions with ice-free drift or not as defined in section 3.1, based on the NSIDC-0051 ice concentration in each hemisphere and different months. As introduced in Lin et al. (2021), the ice vectors from observations and models are removed when ice concentrations are below 50 %, or the data are closer than 75 km to the coast, or with a spurious value.

To be consistent, we apply these selections to other variables in the calculation.

    In the Arctic interior region, the ice-motion MKE in NorESM2-LM/C (Fig. 7a, solid orange) is larger than that in KIMURA (solid blue) and ICDC-NSIDCv4.1 data (solid purple) and this positive bias is slightly reduced in NorESM2-LM/J (solid green) from January to April and September. The largest

improvement in the interior ice-motion MKE occurs in September (Fig. 7b, solid black). It is mostly caused by the increased ice concentration and thickness in NorESM2-LM/J (Figs. 7c to f), while the changes in the surface wind stress being very small (Figs. 7g to h). The September ice concentration in NorESM2-LM/J is close to NSIDC-0051 and OSI-450 data (Fig. 7c). The observational Arctic ice thickness data in September is not available for comparison (Fig. 7e). The ice thickness observations

during 2003-2007 are restricted to a few months per year in both Envisat and ICESat datasets. The Envisat ice thickness data is provided from November to April for the Arctic with coverage up to 81.5$^o$N and May to October for the Antarctic. The measurement campaigns of ICESat ice thickness is for the months of February–March, March–April, May–June, and October–November, with each campaign lasting roughly 33 d. The comparisons between individual models and the two observational

references are thus restricted to these months when data are available (Figs. 7e and 8e). Compared to the OMIP1 simulation, the Arctic interior ice-motion MKE from January to December is not much improved in CMCC-CM2-SR5/J, MRI-ESM2-0/J, EC-Earth3/J and MIROC6/J (Figs. A8a to d). The positive September ice-motion MKE bias in NorESM2-LM/C is larger than in other OMIP1 models, and this positive bias is reduced in NorESM2-LM/J but not in other models.

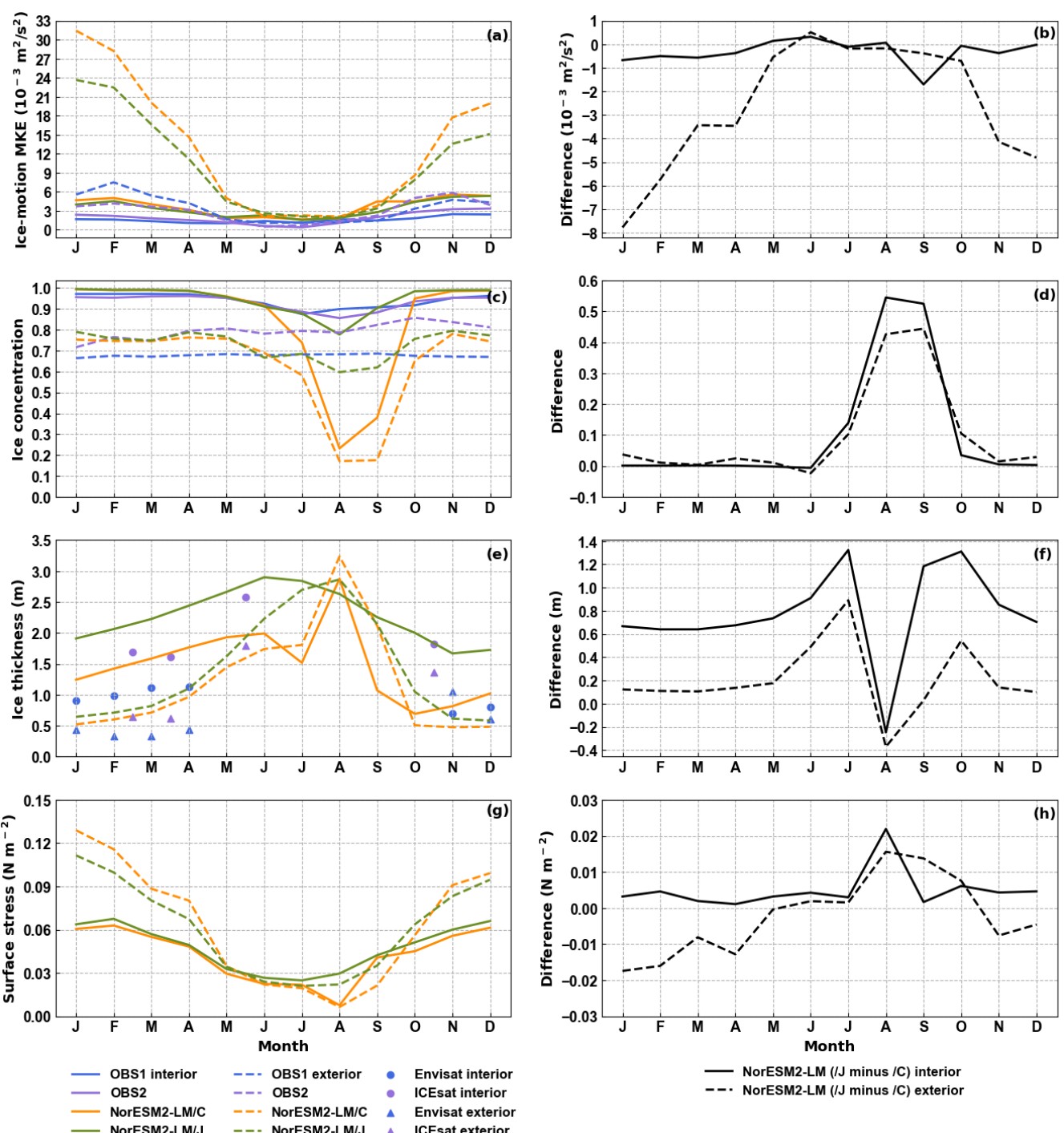

**Figure 7.** 2003-2007 monthly mean and spatially averaged Arctic ice kinetic energy (MKE) (a), ice concentration (c), ice thickness (e) and surface wind stress (g) from observations (blue and purple), NorESM2-LM/C (orange) and NorESM2-LM/J (green). Two observational datasets are included for ice-motion MKE (KIMURA and ICDC-NSIDCv4.1), concentration (NSIDC-0051 and OSI-450) and thickness (Envisat and Icesat). The solid and dashed lines are spatial averages on the regions with ice concentration larger (interior) and smaller (exterior) than 80% in NSIDC-0051, respectively. The differences between NorESM2-LM/J and NorESM2-LM/C ice-motion MKE (b), ice concentration (d), ice thickness (f) and surface wind stress (h) in the interior (solid black) and exterior (dashed black) regions are shown.

In the Arctic exterior region, the ice-motion MKE in the five OMIP1 simulations (Figs. 7a, A8a to d, dashed orange) is much larger than that in KIMURA (dashed blue) and ICDC-NSIDCv4.1 data (dashed

purple) and the positive biases in OMIP1 simulations are largely reduced in OMIP2 simulations (dashed green) from November to April. The decreased Arctic ice-motion MKE in OMIP2 simulations in the exterior region from November to April is mainly induced by the decreased surface wind stress (Figs. 7g and h, A8m and n), while the changes in ice concentration and thickness being very small (Figs. 7c to f and A8e to l). There is no consistent improvement of the representation of sea ice concentration and thickness during November to April from OMIP1 to OMIP2 simulations. We average modeled ice thickness limited up to 81.5°N as in Envisat data and this affects the ice thickness in the summer months but not from November to April. The sea ice thickness is larger in summer than in winter in models (Figs. 7e and A8i to l). As explained before, regions selected to do the spatial average are different in each month in our calculation and this can affect the monthly mean ice thickness. When open water is included in calculating the spatial average of ice thickness, the summer maximum in ice thickness does not exist. The annual maximum and minimum in ice mass are in spring and late summer, respectively. Multi-category sea ice models assume that all sea ice categories melt at the same rate. Consequently, thin ice melts away first and thicker deformed ice remains until late summer. Then, mean ice thickness could be larger in summer than in winter, in particular when total ice concentration is low.

In the Antarctic interior region, the ice-motion MKE in NorESM2-LM/C (Fig. 8a, solid orange) is larger than that in KIMURA (solid blue) and ICDC-NSIDCv4.1 data (solid purple) and this positive bias is reduced in April in NorESM2-LM/J (solid green, smaller than $-1\times10^{-3}$ $m^2/s^2$ as a baseline). The decreased Antarctic ice-motion MKE in the interior region in April (Fig. 7b, solid black) are consistent with the increased ice concentration and thickness but not the increased surface wind stress in NorESM2-LM/J (Figs. 7d, f and h, solid black). However, the ice concentration and thickness are also increased in NorESM2-LM/J from December to March, similarly to April, while the ice-motion MKE is not reduced. This suggests that the ice motion at the beginning of the melting season is not that sensitive to the ice concentration and thickness changes. Olason and Notz (2014) showed that the Arctic ice drift speed in April and May is not correlated with ice concentration or thickness, and the increase in drift speed is due to newly formed fractures without changes in ice concentration. It indicates that the not reduced Antarctic ice-motion MKE from December to March in NorESM2-LM/J can be related to newly formed fractures even though the increases in ice concentration and thickness are similar to that in April. Compared to the OMIP1 simulation, the Antarctic interior ice-motion MKE from January to December is not much improved in CMCC-CM2-SR5/J, MRI-ESM2-0/J, EC-Earth3/J and MIROC6/J (Figs. A9a to d).

In the Antarctic exterior region, the ice-motion MKE in NorESM2-LM/C (Fig. 8a, dashed orange) is larger than that in KIMURA (dashed blue) and ICDC-NSIDCv4.1 data (dashed purple) and this positive bias is reduced in NorESM2-LM/J from July to September (dashed green, smaller than $-1\times10^{-3}$ $m^2/s^2$ as

a baseline). The ice-motion MKE positive bias is reduced from July to October in CMCC-CM2-SR5/J,
MRI-ESM2-0/J, EC-Earth3 and MIROC6 (Figs. A9a to d, dashed black). The reduced ice-motion MKE
in NorESM2-LM/J, CMCC-CM2-SR5/J and MRI-ESM2-0/J is related to the decreased wind stress and
increased ice thickness (Figs. 8f and h, A9i, j, m and n, dashed black). The reduced positive bias of ice-
motion MKE in the Antarctic exterior region is much smaller than that in the Arctic in OMIP2
simulations (Figs. 8b vs. 7b, A9a to d vs. A8a to d, dashed black).

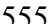

**Figure 8. Same as Fig. 7 but for the Antarctic. The Envisat ice thickness data is provided from May to October.**

### 3.2.2 Ice drift direction and its connections to wind stress

We finally aim at determining to what extent the change in atmospheric forcing may lead to an
560 improvement in the simulated ice drift direction (independently of the improvements in sea ice drift
magnitude noted in the previous section). To that end, the vector correlations between simulated and
observed ice drift fields (KIMURA and ICDC-NSIDCv4.1 data) are diagnosed, as done in Lin et al.
(2021). In general, the vector correlation coefficients between the modeled ice drift and observational
data during 2003–2007 are larger in NorESM2-LM/J than that in NorESM2-LM/C in the Arctic (Figs.
9d and e) and Antarctic (Figs. 9j and k).

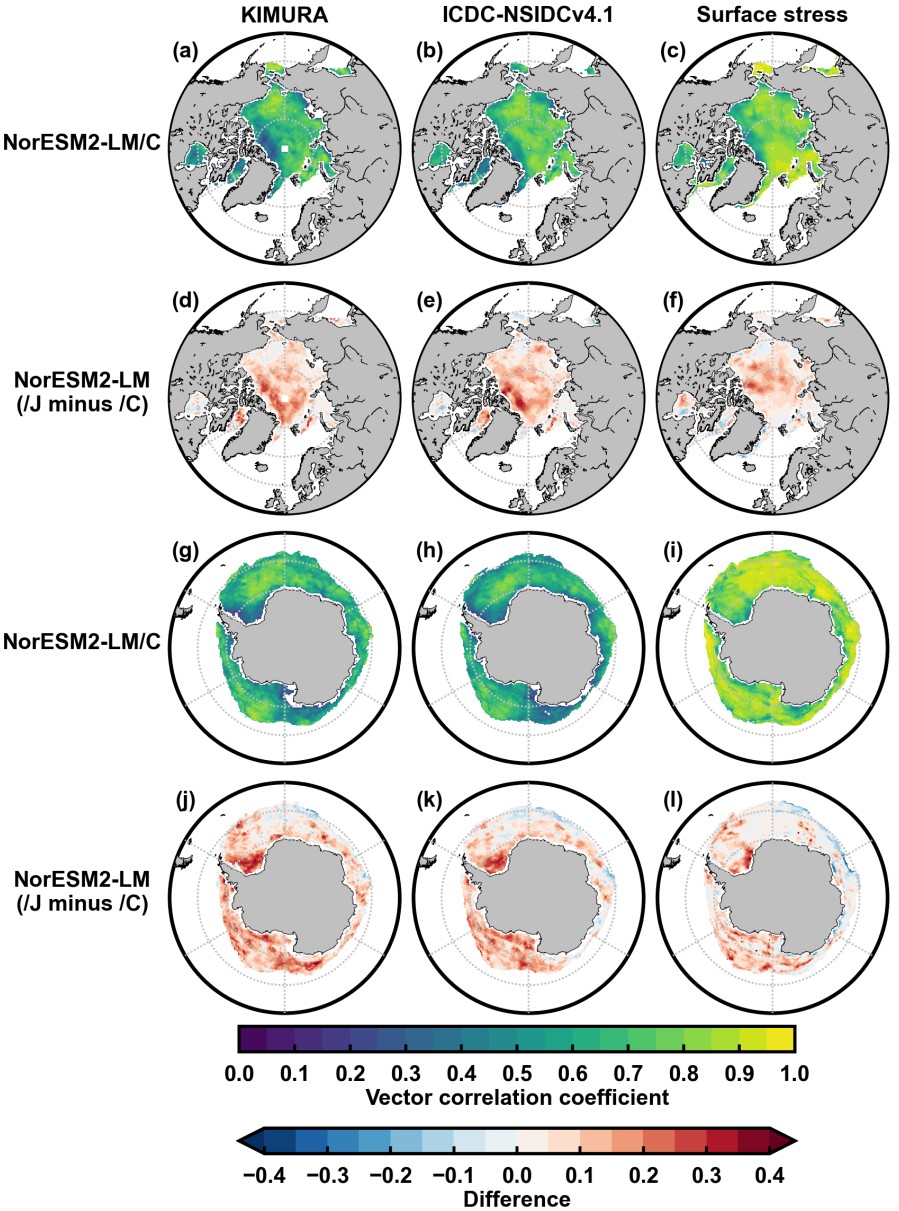

**Figure 9. The significant vector correlation coefficients during 2003–2007 at a level of 99% between modeled ice drift (NorESM2-LM/C) and two observational data (KIMURA and ICDC-NSIDCv4.1), respectively, and between NorESM2-LM/C modeled ice drift and surface wind stress in the Arctic (a to c) and Antarctic (g to i). The second and**
570 **fourth rows are the vector correlation coefficient differences by changing the modeled ice drift from NorESM2-LM/C to NorESM2-LM/J.**

The links with the surface wind stress are assessed. The vector correlation coefficients between modeled ice drift and surface wind stress are much larger in NorESM2-LM/J than that in NorESM2-LM/C in the Beaufort Gyre area (Fig. 9f) and the Pacific and Atlantic sectors of the Southern Ocean (Fig. 9l). Those regions correspond to large improvements in the ice vector direction simulation in NorESM2-LM/J (Figs. 9d, e, j and k). This suggests that the improved ice vector direction simulation is related to the changed surface wind stress in NorESM2-LM/J. These improvements can also be found in CMCC-CM2-SR5/J, MRI-ESM2-0/J, EC-Earth3/J, and MIROC6/J in both hemispheres (Fig. A10), but the improvements in MRI-ESM2-0/J, EC-Earth3/J, and MIROC6/J are smaller than that in NorESM2-LM/J and CMCC-CM2-SR5/J in the Arctic.

## 4 Conclusions and discussion

The OMIP provides useful datasets to reconstruct sea ice evolution over the past decades. Lin et al. (2021) have shown that the accuracy of the reconstruction depends on the atmospheric forcing used. This paper attempts to explain why this is so by conducting surface momentum and heat flux analyses. The two atmospheric reanalysis products are different in both dynamical and thermodynamical components for the Arctic and Antarctic, such as the air temperature and winds, which contribute to heat flux and momentum flux differences in the ocean-sea ice models. We studied the dynamic and thermodynamic processes contributing to the ice concentration tendencies and their links with surface heat and momentum fluxes, as well as the connections between the simulated ice drift and the ice concentration, ice thickness and wind stress.

In general, the sea ice concentration and ice drift magnitude and direction simulations are improved from OMIP1 to OMIP2, and improvements in the Arctic are larger than that in the Antarctic. The net surface heat fluxes are decreased in the interior region with ice concentration above 80% and increased in the CAA and CWS regions during March to August (Arctic) and October to January (Antarctic) in OMIP2 compared to OMIP1 simulations. This can explain the improved OMIP2 ice concentration simulations in the summer, pointing to the important role played by the thermodynamic processes during the ice melting season. The changed net shortwave radiation fluxes from OMIP1 to OMIP2 simulations are crucial to improve the OMIP2 summer ice concentration simulations in the Arctic interior, the CAA and CWS regions. The decreased surface wind stress in the inner part of the exterior region during March to August in OMIP2 compared to OMIP1 contributes to the improved (decreased) Antarctic September OMIP2 ice concentration simulation in the exterior region from 70 to 180°E, pointing at the dominant role of dynamic processes. The monthly mean and spatial averaged Arctic ice-motion MKE simulation in the exterior region is improved (decreased) from November to April due to the decreased surface wind stress in OMIP2 compared to OMIP1 simulations, while the improvement in the Antarctic is small. The improved surface wind stress simulation in the Beaufort Gyre area and the

Pacific and Atlantic sectors of the Southern Ocean can help improve the ice vector direction simulation.


This study provides clues to improve the atmospheric reanalysis products for a better sea ice simulation in ocean-sea ice models. The net shortwave radiation fluxes during the ice melting season in the interior region and the wind stress during the ice expansion season in the exterior region are crucial for a better sea ice concentration simulation. The wind stress is also important to the sea ice drift magnitude and

vector direction simulations. Some aspects of the sea ice simulation are not improved by changing the forcing from CORE-II to JRA55-do, such as the winter Arctic ice concentration in the exterior region, summer Antarctic ice concentration in the coastal regions and ice drift magnitude in some months. The biases in surface heat fluxes and surface stress in these regions and periods are large compared to ERA5 reanalysis data (not shown). Improving Antarctic radiation fluxes, and Arctic and Antarctic winds in the

atmospheric reanalysis products can be helpful to reduce the bias. The limited impact of atmospheric forcing on ice concentration simulation was discussed in Barthélemy et al. (2018). In these exterior and coastal regions, thermodynamic processes tend to compensate for the changes caused by dynamic processes and properly combined contributions from thermodynamic and dynamic processes are needed to improve the simulations. Both atmospheric forcing and model physics of the sea ice growth and melt

processes are crucial for an improved simulation in these aspects. Differences are shown in OMIP models with different sea ice models. For example, the summer sea ice concentration conditions in NorESM2-LM (Fig. 1) and EC-Earth3 (Fig. A2) are different, which can be related to the different radiative schemes in CICE5.1.2 and LIM3 sea ice models as detailed in Table 1. The collaborations with model development groups are needed to help advance the sea ice simulation. We recommend that

in such inter-comparison exercises, all the specificities/namelists of the sea ice models used should systematically be reported to help understand the different responses to model physics.

From our analysis, the differences in the atmospheric forcing are transferred to the modeled surface fluxes and contribute to the improved ice concentration simulation. However, tuning is also a key aspect

in climate models that can explain differences in performance. It is possible that some groups could have tuned for OMIP2 and then used the same setup for OMIP1, so part of the improvement with OMIP2 could be due to this experimental setup choice. While this paper reiterates the importance of the atmospheric forcing for the representation of the sea ice state, it is expected, based on the conclusions, that errors in the atmospheric forcing also affect the ocean through modified heat, freshwater and

momentum fluxes between the ice and the ocean. These errors can thus eventually affect the representation of ocean temperature, salinity and currents.

## Appendix A

In this appendix, extra sea ice diagnostics from CMCC-CM2-SR5, MRI-ESM2-0, EC-Earth3 and MIROC6 are given to help support the conclusions derived from NorESM2-LM. The ice concentration simulations from four models are provided in Figs. A1 and A2. The effects of the thermodynamic and dynamic components of the atmospheric forcing in CMCC-CM2-SR5 and MRI-ESM2-0 are presented in Figs. A3 to A6. The surface heat flux on sea ice is not provided for MRI-ESM2-0 and the corresponding figures are not included in Figs. A5 and A6. The downward and upward shortwave radiation flux in NorESM2-LM and CMCC-CM2-SR5 are added in Fig. A7. The ice drift simulation and the relationship to ice concentration, ice thickness and wind stress in CMCC-CM2-SR5, MRI-ESM2-0, EC-Earth3 and MIROC6 are provided in Figs. A8 to A10.

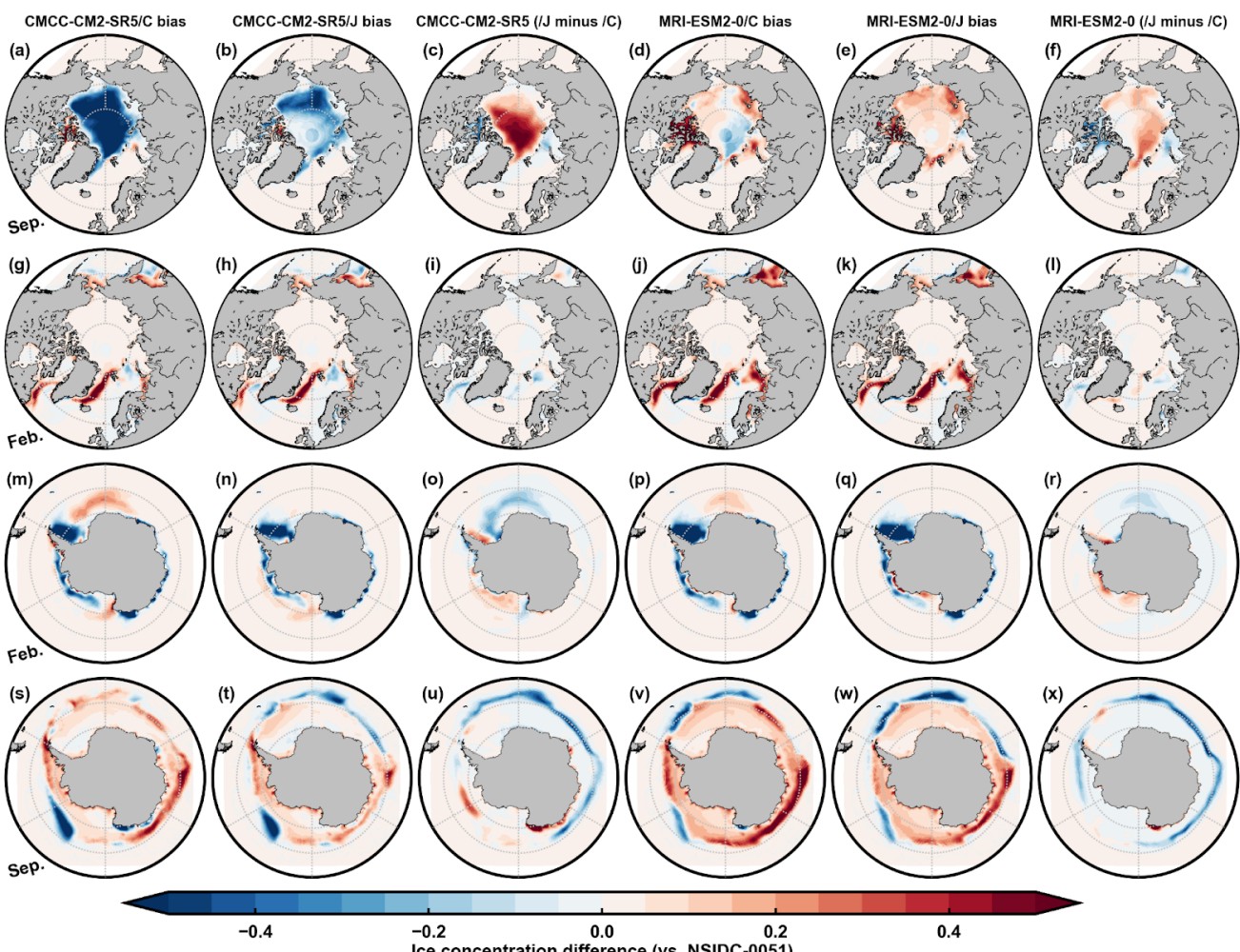

**Figure A1. 1980-2007 September and February mean Arctic (a to l) and Antarctic (m to x) sea ice concentration differences between CMCC-CM2-SR5/C and NSIDC-0051 (first column), CMCC-CM2-SR5/J and NSIDC-0051 (second column), CMCC-CM2-SR5/J and CMCC-CM2-SR5/C (third column), MRI-ESM2-0/C and NSIDC-0051 (fourth column), MRI-ESM2-0/J and NSIDC-0051 (fifth column), and MRI-ESM2-0/J and MRI-ESM2-0/C (sixth column).**

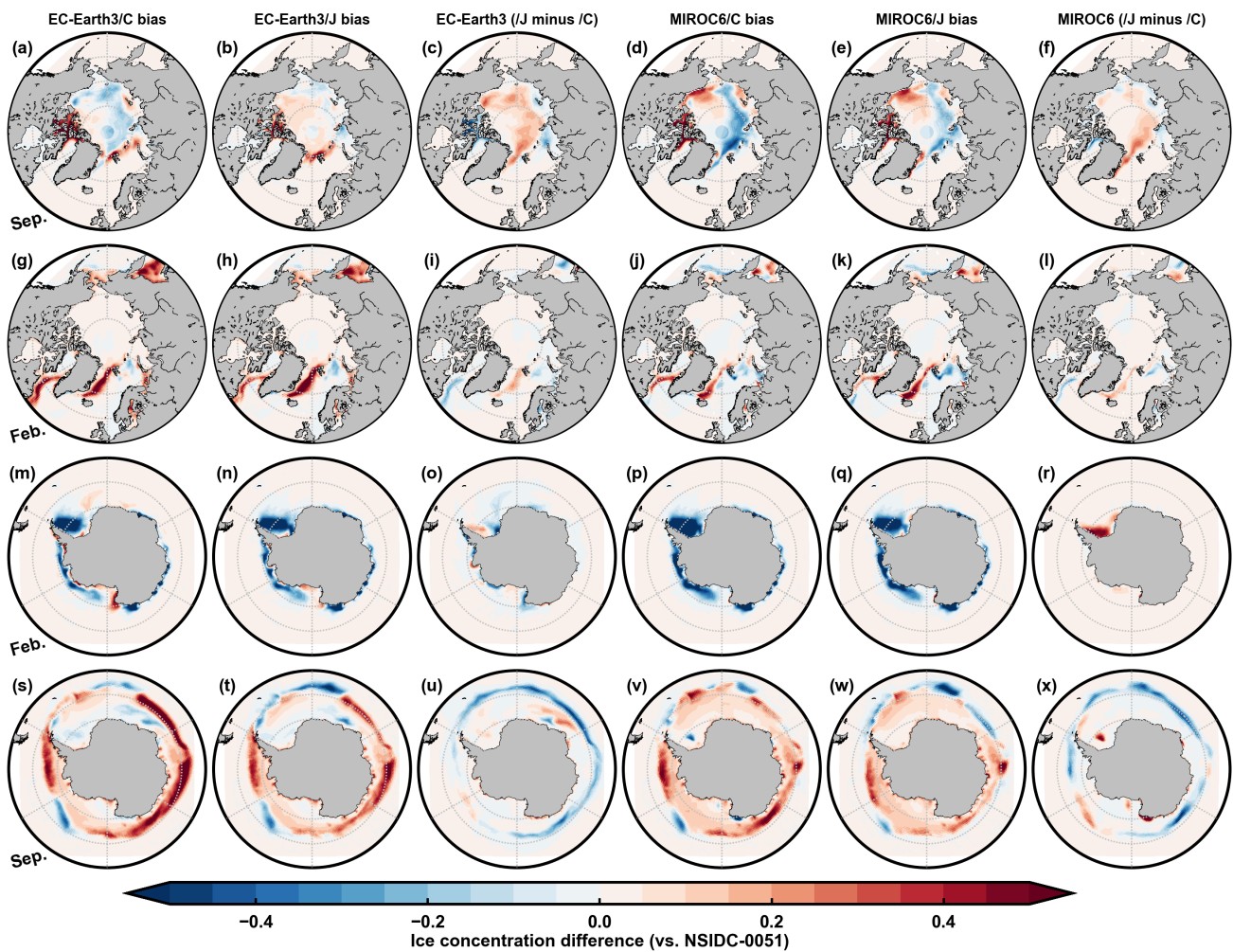


**Figure A2.** 1980-2007 September and February mean Arctic (a to l) and Antarctic (m to x) sea ice concentration differences between EC-Earth3/C and NSIDC-0051 (first column), EC-Earth3/J and NSIDC-0051 (second column), EC-Earth3/J and EC-Earth3/C (third column), MIROC6/C and NSIDC-0051 (fourth column), MIROC6/J and NSIDC-0051 (fifth column), and MIROC6/J and MIROC6/C (sixth column).


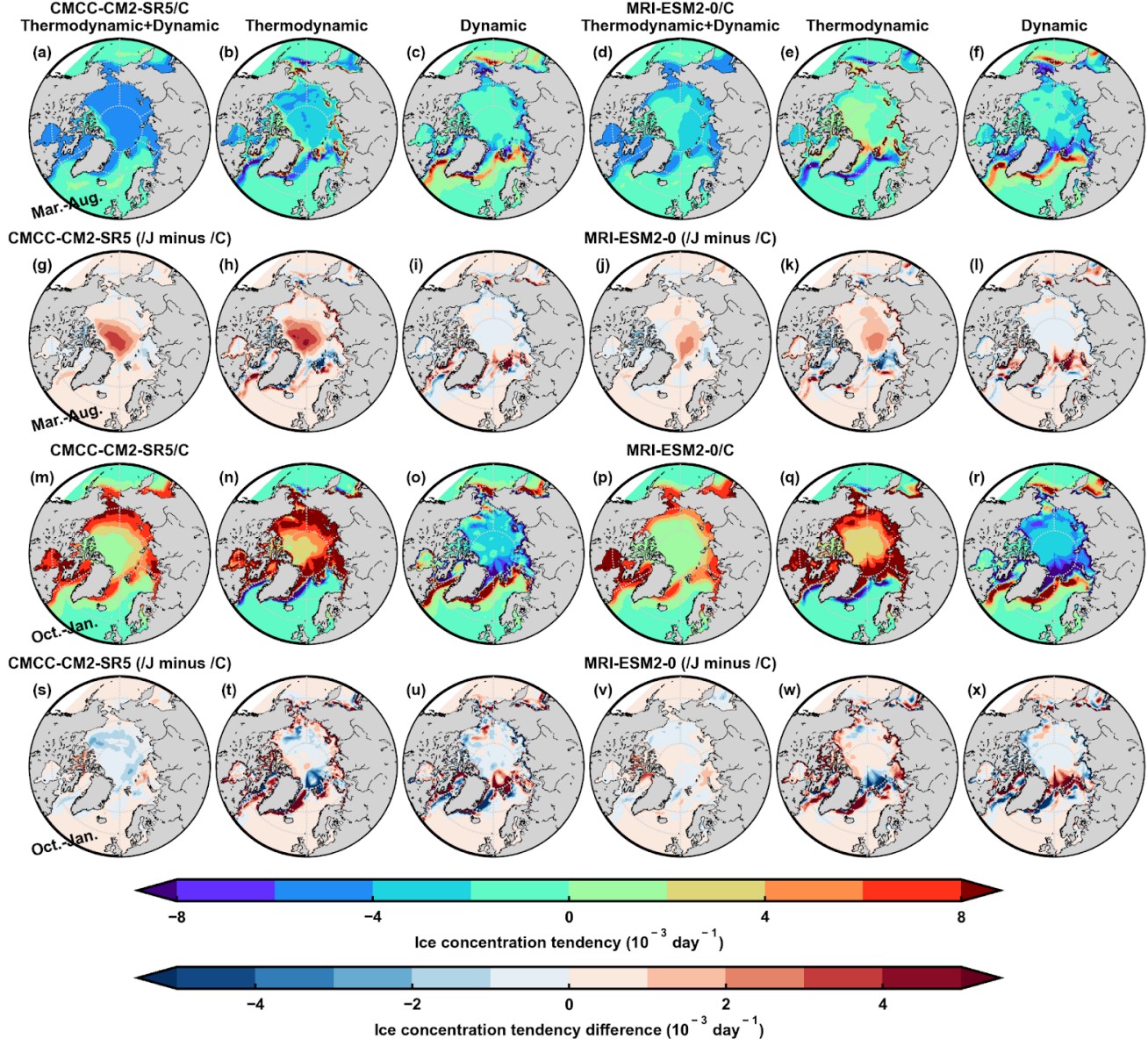

**Figure A3. 1980-2007 March-August (a to l) and October-January (m to x) mean Arctic sea ice concentration tendencies in CMCC-CM2-SR5 (first three columns) and MRI-ESM2-0 (last three columns) due to thermodynamic and dynamic processes in total (first and fourth columns), thermodynamic processes (second and fifth columns) and dynamic processes (third and sixth columns). The first and third rows are from CMCC-CM2-SR5/C and MRI-ESM2-0/C, and the second and fourth rows are differences between CMCC-CM2-SR5/J and CMCC-CM2-SR5/C, and between MRI-ESM2-0/J and MRI-ESM2-0/C, respectively.**


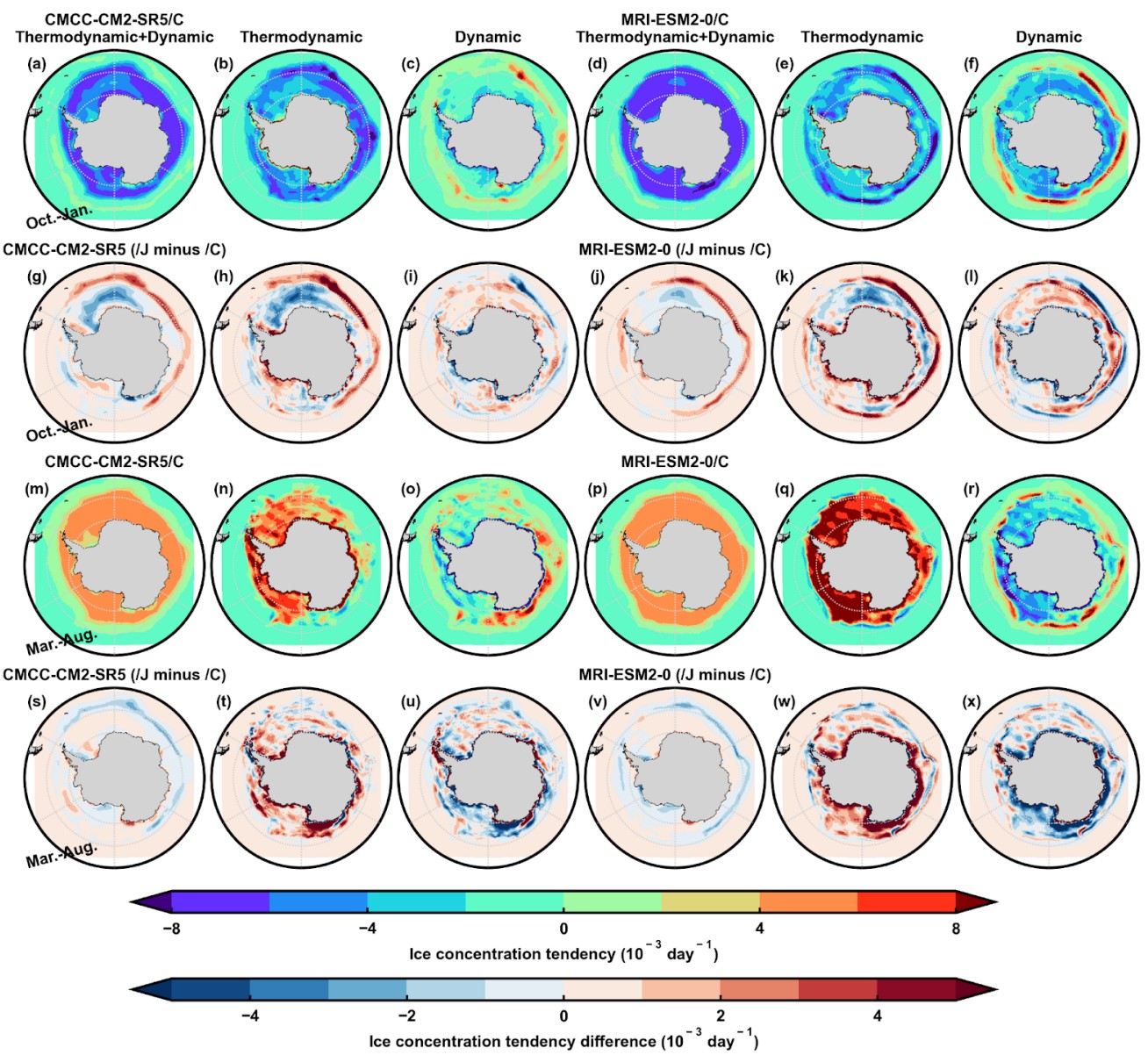

Figure A4. Same as Fig. A3 but for the 1980-2007 October-January (a to l) and March-August (m to x) mean Antarctic sea ice concentration tendencies.

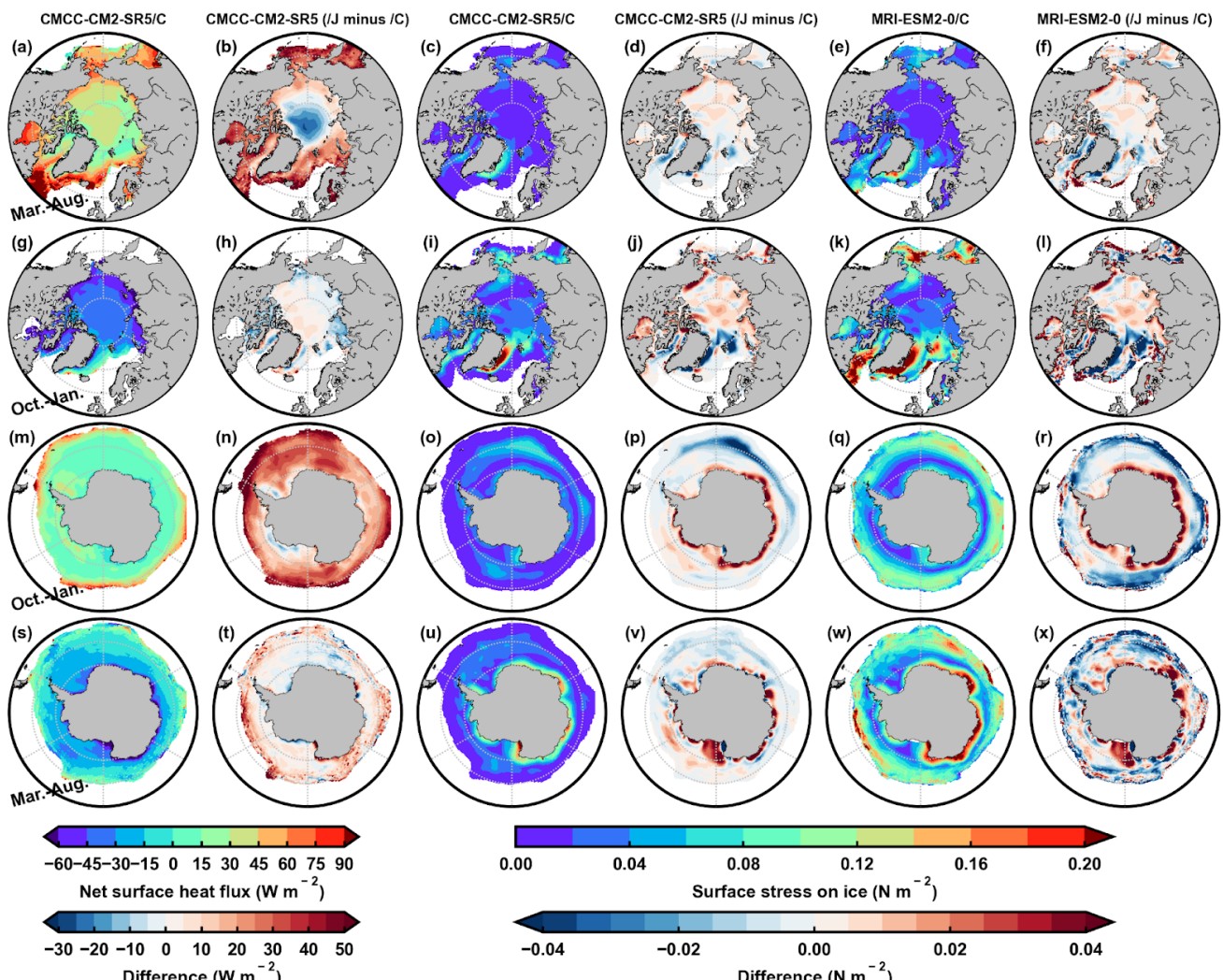

**Figure A5. 1980-2007 March-August and October-January mean Arctic (a to l) and Antarctic (m to x) net surface heat flux (first two columns) and surface stress (last four columns) on sea ice. The positive values indicate a surface flux downward. The first and third columns correspond to CMCC-CM2-SR5/C, and the second and fourth columns are differences between CMCC-CM2-SR5/J and CMCC-CM2-SR5/C. The fifth column corresponds to MRI-ESM2-0/C, and the sixth column is the difference between MRI-ESM2-0/J and MRI-ESM2-0/C.**


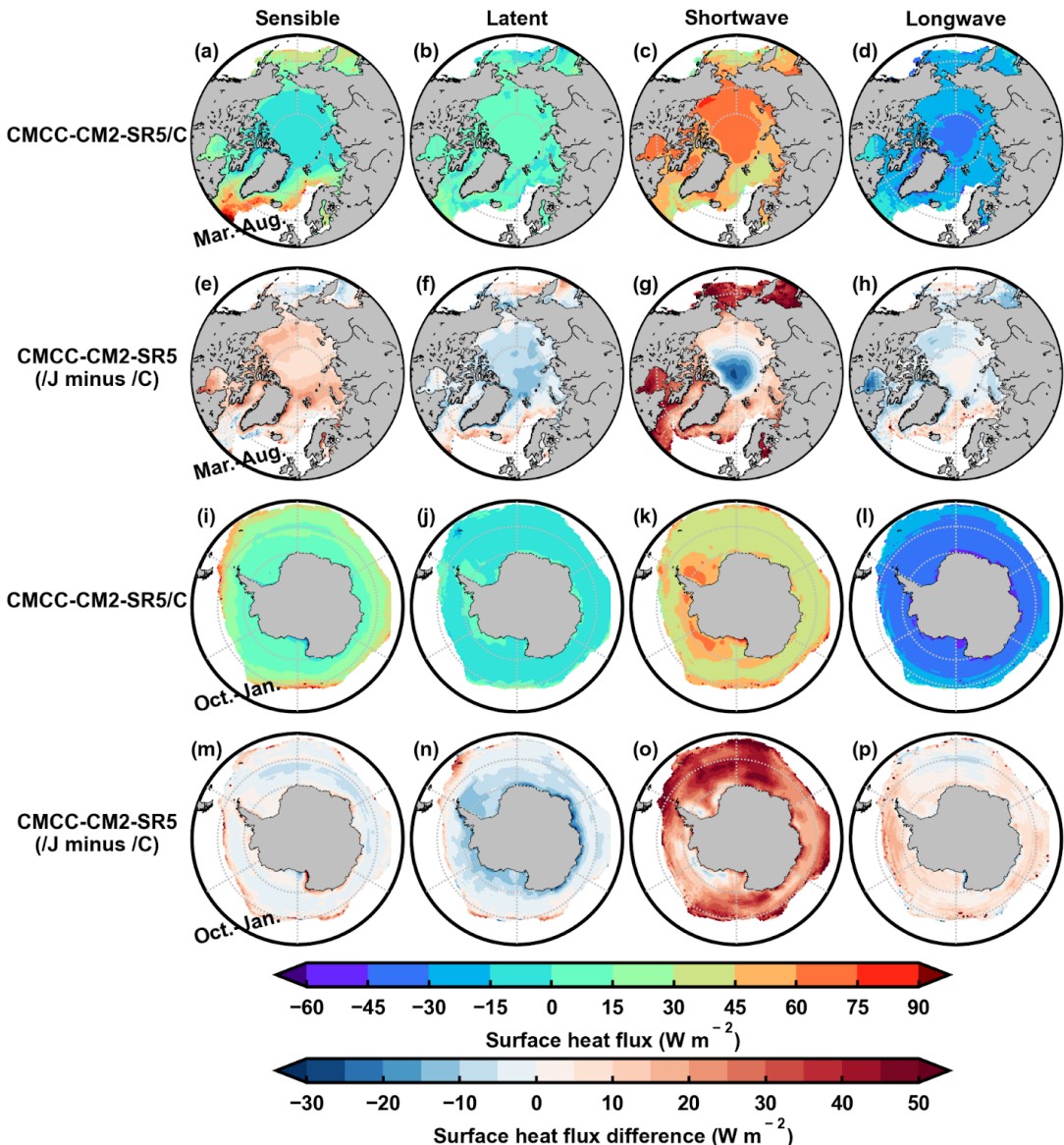

**Figure A6.** 1980-2007 March-August mean Arctic (a to h) and October-January mean Antarctic (i to p) surface sensible (first column) and latent heat fluxes (second column), net shortwave (third column) and longwave radiation fluxes (fourth column). The positive values indicate a surface flux downward. The first and third rows correspond to CMCC-CM2-SR5/C, and the second and fourth rows are differences between CMCC-CM2-SR5/J and CMCC-CM2-SR5/C.

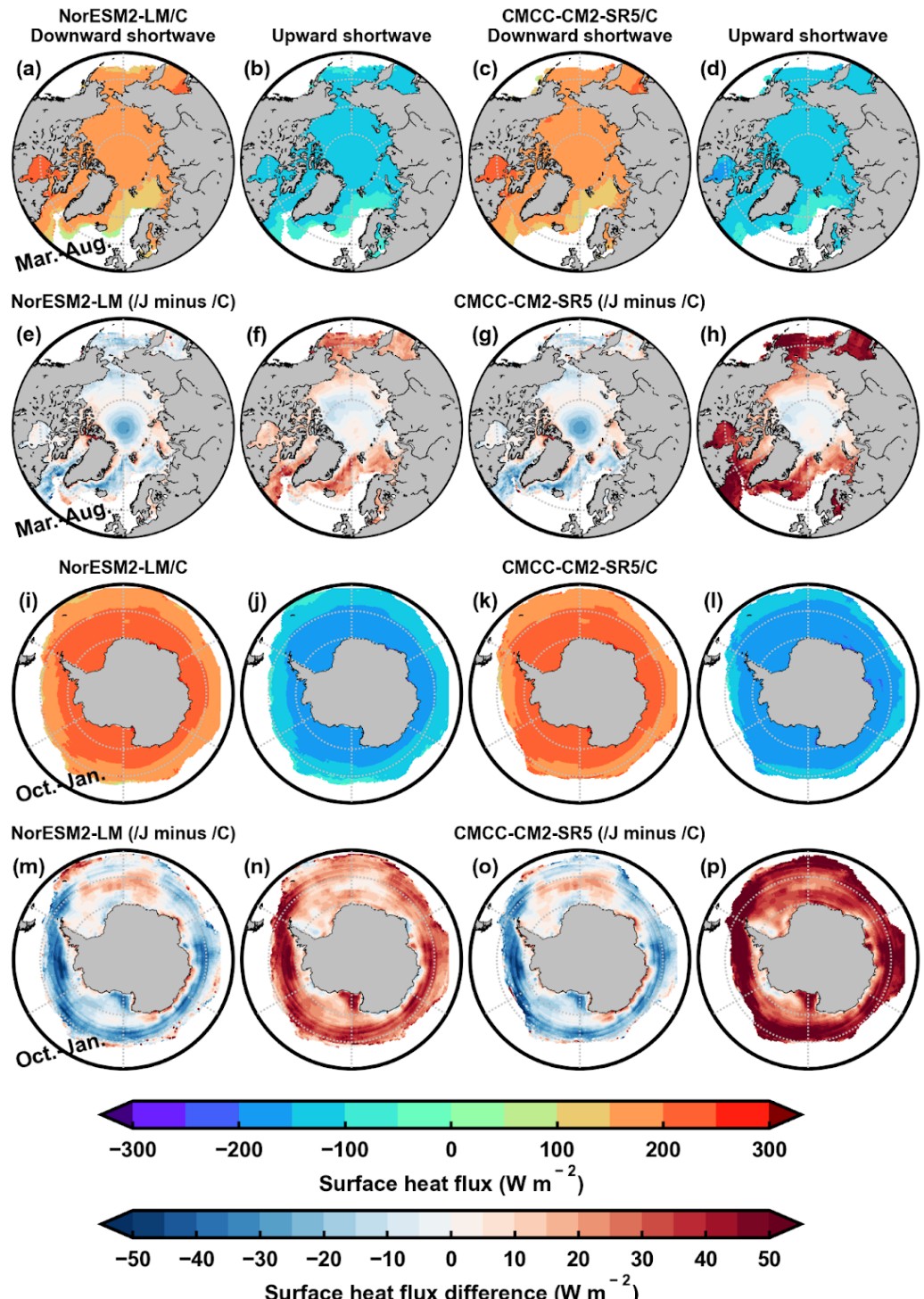

**Figure A7.** 1980-2007 March-August mean Arctic (a to h) and October-January mean Antarctic (i to p) downward and upward shortwave radiation fluxes in NorESM2-LM (first two columns) and CMCC-CM2-SR5 (last two columns). The positive values indicate a surface flux downward. The first and third rows correspond to model/C, and the second and fourth rows are differences between model/J and model/C.

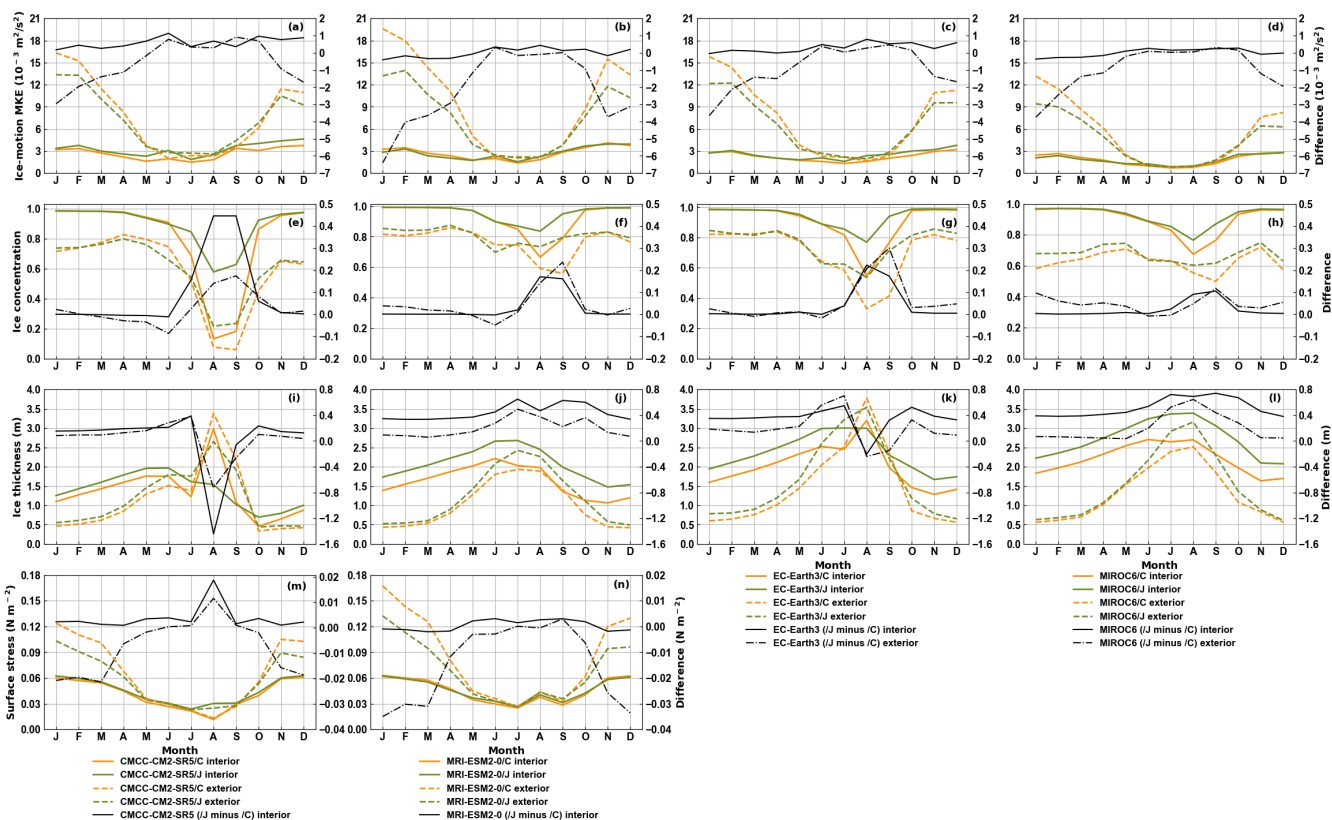

Figure A8. 2003-2007 monthly mean and spatially averaged Arctic ice kinetic energy (MKE) (a to d), ice concentration (e to h), ice thickness (i to l) and surface wind stress (m and n) in model/C (orange), model/J (green), and differences between model/J and model/C (black). The first to fourth columns correspond to CMCC-CM2-SR5, MRI-ESM2-0, EC-Earth3, MIROC6 model results, respectively. The solid and dashed lines are spatial averages on the regions with ice concentration larger (interior) and smaller (exterior) than 80% in NSIDC-0051, respectively.

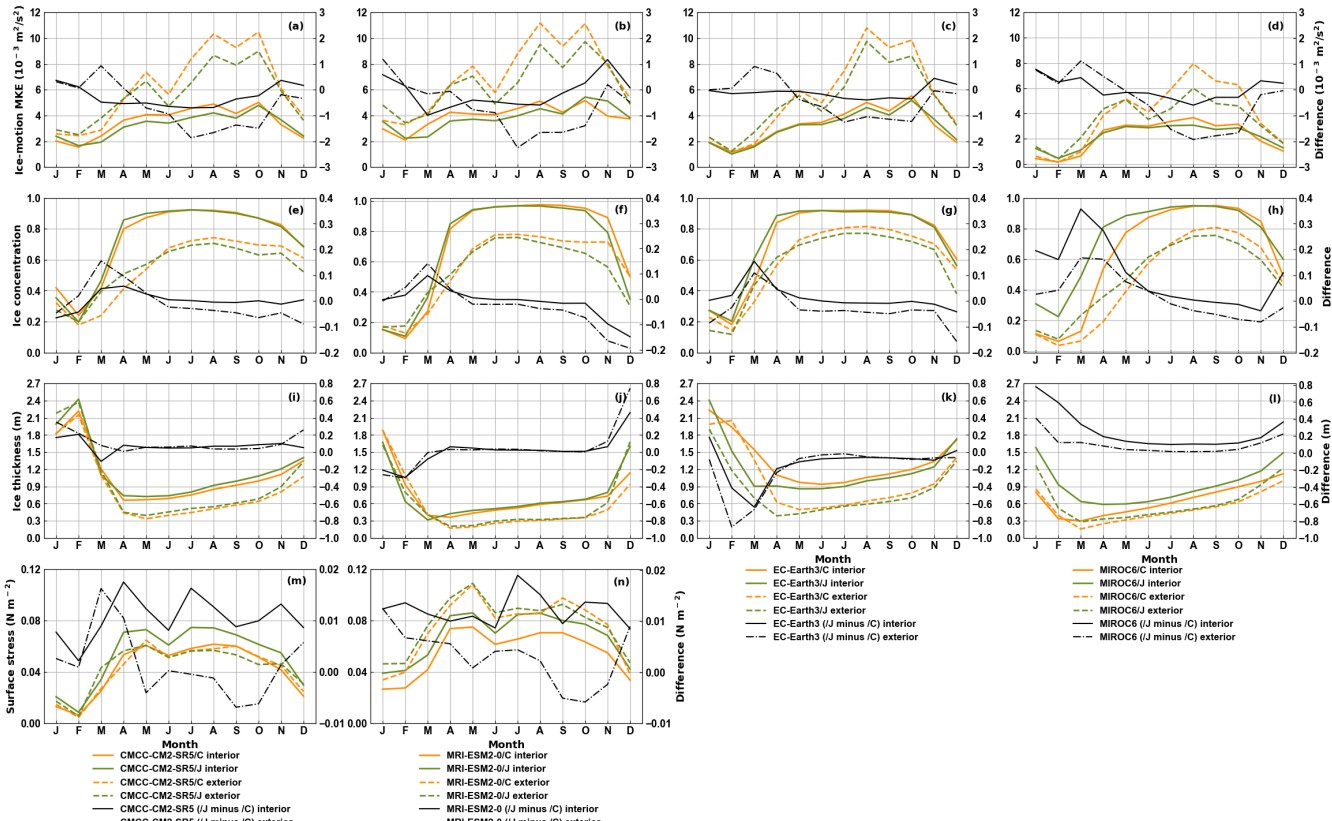

**Figure A9. Same as Fig. A8 but for the Antarctic.**

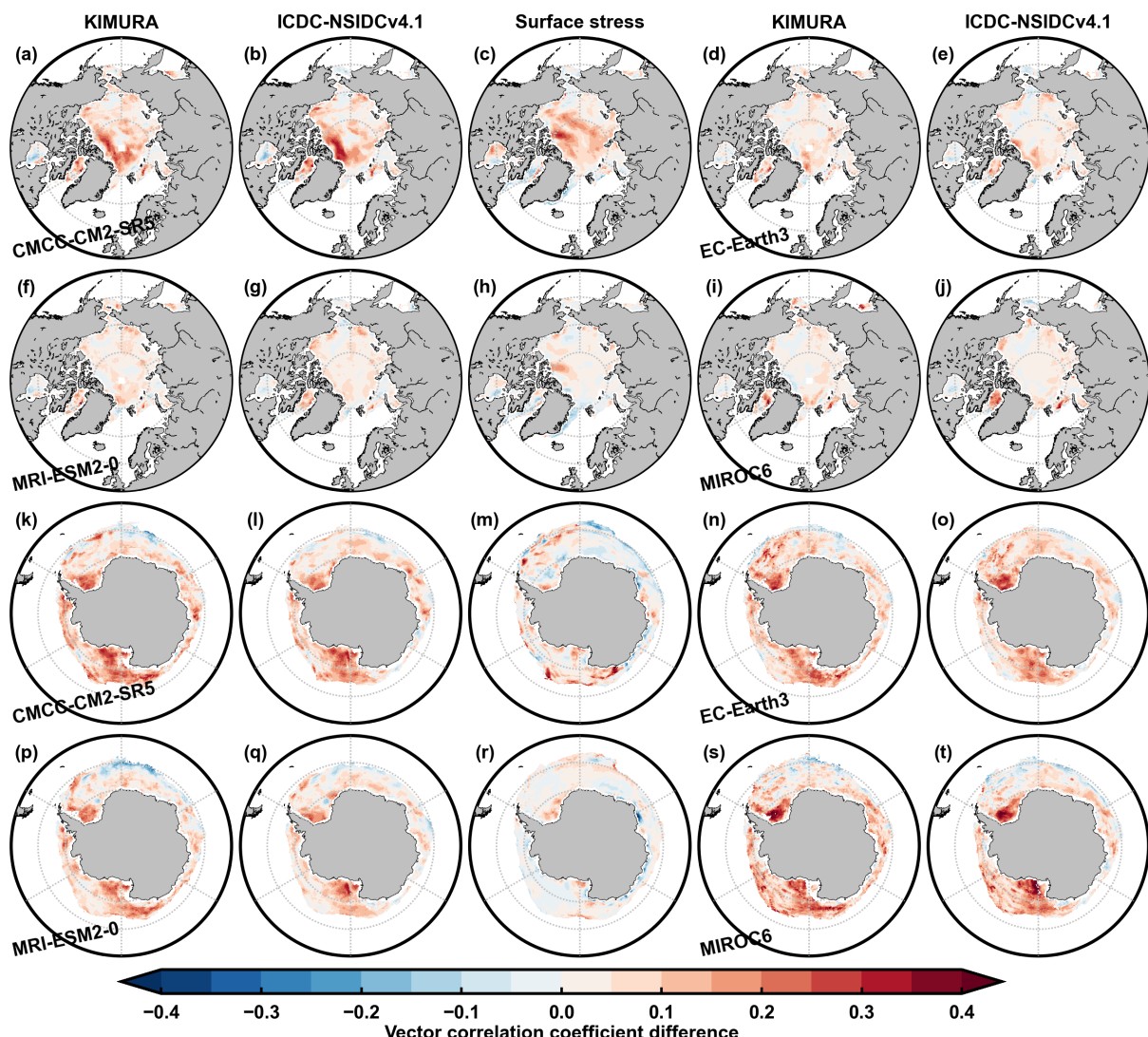

Figure A10. Differences of significant vector correlation coefficients during 2003–2007 at a level of 99% between model/J and model/C in the Arctic (a to j) and Antarctic (k to t). The first, second, fourth and fifth columns are significant vector correlation coefficients between modeled ice drift and KIMURA/ICDC-NSIDCv4.1 data, and the third column are significant vector correlation coefficients between modeled (CMCC-CM2-SR5 and MRI-ESM2-0) ice drift and surface stress.

*Data availability.* CMIP6 OMIP data are freely available from the Earth System Grid Federation. The download links to the observational references used in this paper are listed in Table 3 of Lin et al.

(2021). The CORE-II data is available at https://data1.gfdl.noaa.gov/nomads/forms/core/COREv2.html and the JRA5-do data is available at https://esgf-node.llnl.gov/search/input4mips (last access: 3 April 2023).

*Author contributions.* XL and FM developed the concept of the paper. XL performed the analysis and led the writing of the paper. All authors contributed to the discussion of the study and the editing of the

manuscript.

*Competing interests.* The authors declare that they have no conflict of interest.

*Acknowledgments.* We are grateful to Noriaki Kimura and Sara Fleury for providing and introducing the ice drift and ice thickness datasets, respectively. We thank the sea ice observational groups and the

climate modelling groups for producing and making available their output. Xia Lin is a F.R.S.-FNRS scientific collaborator. François Massonnet is a F.R.S.-FNRS research fellow.

*Financial support.* This research has been supported by the Copernicus Marine Environment Monitoring Service (CMEMS) SI3 project. CMEMS is implemented by Mercator Ocean International

in the framework of a delegation agreement with the European Union. Xia Lin also received support from the National Natural Science Foundation of China (grant nos. 41941007, 41906190, and 41876220) and the Innovation Group Project of Southern Marine Science and Engineering Guangdong Laboratory (Zhuhai) (grant no. 311021008).

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
