# Peer review of "Impact of atmospheric forcing uncertainties on Arctic and Antarctic sea ice simulations in CMIP6 OMIP"

_The Cryosphere, 2022_

## Referee Comment (RC1)

Review for:
**Impact of atmospheric forcing uncertainties on Arctic and Antarctic**

**sea ice simulation in CMIP6 OMIP**

Xia Lin, François Massonnet, Thierry Fichefet, Martin Vancoppenolle;
https://doi.org/10.5194/tc-2022-110

The manuscript "Impact of atmospheric forcing uncertainties on Arctic and Antarctic sea ice simulation in CMIP6 OMIP" by Lin et al. 2022 presents a study, that addresses the impact of different atmospheric forcing on the CMIP6 Ocean Model Intercomparison Project (OMIP). The paper focuses on the simulated sea ice condition by three OMIP models that provide sea ice tendencies for a complete analysis of the results.

The study is well structured and organized. It addresses the newest model versions of model intercomparison project, the CMIP6. Methods and data are appropriate, and the results are clearly presented and described (text and figures). The referencing is good. The paper presents a useful report of the model experiments with different forcing datasets, but could profit from more ambitious interpretation of the physical processing behind the results, and evaluation of the atmospheric results against reference datasets. Here I suggest three main points that the authors could consider to further improve the manuscript:

- Comparison of the forcing datasets (COREII and JRA55-do)
    - You could compare the surface energy fluxes and the near surface wind speeds of the two input datasets. This would allow to say how directly the differences in the input data are transferred to the model results
- More focus on the evaluation and interpretation of the energy balance terms
    - Are the model results realistic, are OMIP2 results better that OMIP1? You could compare the modeled fluxes against literature values or reference data. I understand that finding a good reference dataset for polar energy fluxes is hard. Still, it would be possible to compare the fluxes against reanalyses (eg. ERA5, JRA55 or NCEP-CFSR) to get an idea of the possible biases. Another approach would be to compare each model against the multimodel mean (mean of CMCC-CM2-SR5, MRI-ESM2-0 and NorESM2-LM), assuming that the multimodel mean is better than each individual model.
- Evaluation of the sea ice thickness
    - There are substantial biases in the sea ice thickness. Can you comment them in any way? Is the reference data reliable (see also my comment for Page 17, lines 399-400), can the limitations in model physics make the ice too thick, what in the forcing data or initial conditions could cause the ice to be too thick?

These improvements could make the paper even more relevant to the audience and turn the "clues on how improved atmospheric reanalysis products influence sea ice simulations" to more than just clues. However, even in the current form, the paper points out the aspects where more research is needed, which is also very relevant for the community. All in all, this a

meticulously prepared manuscript, where the expertise of the authors is visible. Thank you for the interesting reading!

I provide some more detailed comments below.

Page 3, Line 87: In this paragraph, you could summarize some main links between the atmospheric circulation and sea ice conditions. You could describe the role of atmospheric circulation and it's impacts on sea ice on a general level. You could mention that some circulation patters allow heat and moisture transport to the Arctic, while others prohibit it.

Page 5, Lines 165-166: To help the reader, you could mention that SIC was overestimated in CCA in OMIP1 and that overestimation was reduced in OMIP2.

Page 8, Table 1: The caption states: "The improvements on ice concentration simulations and the related reasons in summer (bold) and winter (bold italic) are marked". Bold and italic text is used also for the sea ice concentration tendencies, the surface heat flux, and the surface stress on sea ice. Is this correct, or should the improvements be marked only for the SIC?

Page 13, line 319: You wrote "downward net shortwave radiation flux" do you mean the net shortwave or the downward shorwave?

Page 13, line 323: You wrote "downward net shortwave radiation flux" do you mean the net shortwave or the downward shorwave?

Page 13, line 326: You wrote "downward net shortwave radiation flux" do you mean the net shortwave or the downward shorwave?

Page 16, Table 2: To me, it would be more intuitive to see the OMIP1 (C) and OMIP2 (J) values instead of C and J-C. Best would be to have all three values (C, J, and J-C), but it's hard to fit them in one table. Also, I would suggest adding the downward longwave and upward longwave radiation, as you did for the shortwave radiation. You will see if the decreased downward shortwave radiation coincides with increased downward longwave radiation. This would suggest an increase in atmospheric moisture/cloudiness between the two experiments.

Page 17, line 390: Could you add a definition or equation for the MKE?

Page 17, lines 399-400: Could you select the modeled sea ice thickness for a domain that matches the Envisat data (< 81.5°N)? This would allow a better comparison between the datasets.

Page 17, line 402: Should "ice drift speed" be "ice-motion MKE"?

Page 17, lines 404-405: Should "ice drift speed" be "ice-motion MKE"?

Page 18, Figure 6: The interior-bar in June is all green. I think that the blue and orange bars might be missing/hidden.

Page 19, line 434: "drift speeds" or "MKE"?

Page 22, line 515: "More attention needs to be paid to the radiation fluxes and wind stress in the atmospheric reanalysis products." Here you could elaborate on what kind of attention is needed. What are the sources of the current problems, what caused the improvements when passing from CORE-II to JRA55-do etc.

---

## Author Comment (AC1)

**Dear reviewer,**

We thank you for the constructive comments on the earlier version of the manuscript. We have revised our manuscript following the comments and our response is as follows.

**Reviewer 1:**

The manuscript "Impact of atmospheric forcing uncertainties on Arctic and Antarctic sea ice simulation in CMIP6 OMIP" by Lin et al. 2022 presents a study, that addresses the impact of different atmospheric forcing on the CMIP6 Ocean Model Intercomparison Project (OMIP). The paper focuses on the simulated sea ice condition by three OMIP models that provide sea ice tendencies for a complete analysis of the results.

The study is well structured and organized. It addresses the newest model versions of model intercomparison project, the CMIP6. Methods and data are appropriate, and the results are clearly presented and described (text and figures). The referencing is good. The paper presents a useful report of the model experiments with different forcing datasets, but could profit from more ambitious interpretation of the physical processing behind the results, and evaluation of the atmospheric results against reference datasets.

**Thanks for your time and positive evaluation.**

Here I suggest three main points that the authors could consider to further improve the manuscript:

- Comparison of the forcing datasets (COREII and JRA55-do)
  - You could compare the surface energy fluxes and the near surface wind speeds of the two input datasets. This would allow to say how directly the differences in the input data are transferred to the model results

**Answer:** Thanks for your suggestion. The surface air temperature, specific humidity, downward shortwave and longwave radiation fluxes during melting months, and wind speed during freezing months in COREII and JRA55-do are shown in Fig. 1 below. These months are shown because in general the ice concentration simulations are improved from OMIP1 to OMIP2 in summer due to surface heat flux changes and in winter due to the wind stress changes. More information can be found in Figs. 1 to 4, A1 to A4 of our manuscript. Compared to COREII, the downward shortwave radiation flux and specific humidity in JRA5-do in the central Arctic Ocean and the coastal region of the western Weddell Sea (Figs. 1g, h, q, r) are lower, the downward shortwave radiation flux in the Canadian Arctic Archipelago (CAA) and central Weddell Sea (CWS) regions and the air temperature in the CAA region are larger (Figs. 1h, r, f), and the surface wind speed on Antarctic sea ice in the inner part of the exterior region from 70° to 180°E is weaker (Figs. 1t). These differences in the atmospheric forcing are transferred to the modeled surface fluxes and contribute to

the improved ice concentration simulation in those regions. Compared to OMIP1 simulations, the downward shortwave radiation flux and latent heat flux in OMIP2 in the central Arctic Ocean and the coastal region of the western Weddell Sea are lower, the downward shortwave radiation flux in the CAA and CWS regions and the sensible heat flux in the CAA region are larger, and the surface wind stress on Antarctic sea ice in the inner part of the exterior region from 70° to 180°E is weaker (Figs. 4, 5, A4, A5, A6 in the previous manuscript).

Action: We have added Fig. 1 below to the revised manuscript as a new Fig. 6 and added the explanation before on how the differences in the atmospheric forcings are transferred to the model results in the main text.

---

## Author Comment (AC3)

Dear reviewer,

We thank you for the constructive comments on the earlier version of the manuscript. We have revised our manuscript following the comments and our response is as follows.

**Reviewer 2:**

First, I apologize to the editor and authors for the delay in writing this review.

The study by Lin et al. investigates an important aspect of stand-alone ocean and sea ice models, which is the relation between the simulated fields and the atmospheric forcings. In simple terms, the paper tries to assess whether a new and arguably better forcing (JRA-55-do) leads to better simulation compared to an older forcing (CORE2). I find this a very interesting question, with important implications also for fully coupled model setups, and I would like to see more studies on these technical but rather important aspects. Congratulations to the authors for pursuing such an interesting problem.

That said, I found the authors' methodological approach unsatisfactory in providing a convincing answer to the problem. The analyses presented here are formally correct, but several central aspects have been almost completely neglected, as illustrated in my comments below. Otherwise, the paper is very well written and structured, but I have some suggestions for improving figures and tables, which sometimes are not adequate.

In summary, I have several major concerns that in my opinion should be addressed before this manuscript is considered for publication. I hope the authors find these helpful for improving their study.

MAJOR COMMENTS:

My biggest concern is that this manuscript does not consider the tuning of the systems analyzed. Let me start by acknowledging that finding out specific details about namelist parameters and other technical information is not trivial when dealing with CMIP-type simulations. Nevertheless, this aspect is important for a study of this kind and cannot be neglected. Specifically, I am wondering whether the CORE2 and JRA-55-do simulations were run with the same model setup, or if the model was specifically tuned for a certain forcing. In my view, tuning is a fundamental step that, given the under-constrained nature and spatiotemporal variability of the sea ice model parameters, must be performed to accommodate a model configuration to a specific forcing. I would argue that the best setup for a study like this would be one where each simulation is optimized to obtain the best compatibility with a set of observations, given a specific atmospheric forcing (i.e., different parameters for CORE2 and JRA-55-do). If this is not the case and an identical model setup is adopted for all atmospheric forcings, we should at least be made aware of whether the model parameters

have been tuned under CORE2 or JRA-55-do, if tuned at all for an OMIP setup. For example, if a model configuration has been tuned under JRA-55 and this same configuration is then run under CORE2, it is not surprising that one would outperform the other. Based on the information currently in the paper, we cannot say anything regarding the previous considerations, which, in my view, is a substantial limitation that should be addressed to pursue the research question from the right angle.

**Answer:** Thanks for your comment. We recognize that tuning is a key aspect in climate models. The design of the CMIP6 OMIP simulations has been organized by the World Climate Research Programme (WCRP) Climate Variability and Predictability (CLIVAR) Working Group on Ocean Model Development Panel (OMDP), and ongoing research collaboration is done through the OMDP to develop OMIP2 (Griffies et al., 2016). Importantly, and to our understanding, the same configuration is used under two different atmospheric forcing datasets as mentioned in Tsujino et al. (2020). It is possible that some groups could have tuned for OMIP2 and then used the same setup for OMIP1, so part of the improvement with OMIP2 could be due to this experimental setup choice. From our analysis, the differences in the atmospheric forcing are transferred to the modeled surface fluxes and contribute to the improved ice concentration simulation (see our response to major comment 4, Fig. 5 below). Thus, the OMIP1 and OMIP2 datasets allow singling out how atmospheric forcing differences are translated into simulated sea ice differences.

**Action:** The information has been added in the revised introduction when introducing the OMIP models. We also added one sentence in the discussion part to address the possible effects from tuning.

I think the manuscript lacks an in-depth description of the model components used in the three systems considered, which could be helpful for a more detailed interpretation of the results. The only information available in Lin et. al. 2021 is that two systems employ different versions of CICE as sea ice model components (CMCC-CM2 and NorESM2). By digging in the MRI-ESM2 model description paper we discover the sea ice component of MRI. COM4.4 is also based on CICE. Given the modularity of CICE, the model version tells us nothing about the physical configuration used by each modeling center. A better description of the model configurations would allow linking differences in the model response to the reanalyses, to differences in the specific physics used. For example, I suspect that different radiative schemes lead to very different sea ice concentration conditions in summer.

**Answer:** We provided some information on the sea ice models in Table 1. For the MRI-ESM2-0 model, the sea ice component is MRI.COM4.4 and the thermodynamics are based on Mellor and Kantha (1989), which are different from the thermodynamics in CICE (Bitz and Lipscomb, 1999). We have added the EC-Earth3 and the MIROC6 models in the revised manuscript (see our response to the next comment, Figs. 1 to 4) and they are based on the LIM3 and COCO4.9 sea ice models, respectively. The radiative schemes in EC-Earth3 and NorESM2-LM are different, which can affect the summer sea ice concentration conditions in EC-Earth3

(Fig. 1) and NorESM2-LM (Fig. 6 below). We cannot have much more information on the radiative schemes of MIROC6 and MRI-ESM2-0.

**Action:** This sea ice model information has been added to the revised section 2. We have added a sentence in the manuscript recommending that, in such inter-comparison exercises, all the specificities/namelists of the sea ice models used should systematically be reported. These comments have been added in the discussion section.

Table 1. The details of five CMIP6-OMIP sea ice models evaluated in the study.

| Model | Sea Ice Model | Sea Ice Component | References |
|-------|---------------|-------------------|------------|
| CMCC-CM2-SR5 | CICE4 | -Energy-conserving thermodynamics;
-Elastic-Viscous-Plastic (EVP) rheology;
-Ice Thickness Distribution (ITD) with five thickness categories; 1 layer of snow and 4 layers of ice;
-A Delta-Eddington multiple-scattering shortwave radiation treatment | Cherchi et al. (2019) |
| EC-Earth3 | LIM3 | -Energy-conserving halo-thermodynamics;
-2-D EVP;
-ITD with five thickness categories; 1 layer of snow and 2 layers of ice;
-The impact of melt ponds is implicitly accounted for through imposed changes on the albedo activated when the surface temperature is $0^o$C. | Döscher et al. (2022) |
| MIROC6 | COCO4.9 | -Energy-conserving thermodynamics on only one layer for sea ice;
-EVP;
-ITD with five thickness categories; | Tatebe et al. (2019) |
| MRI-ESM2-0 | MRI.COM4.4 | -Energy-conserving thermodynamics based on Mellor and Kantha (1989);
-EVP;
-ITD with five thickness categories; | Yukimoto et al. (2019) |
| NorESM2-LM | CICE5.1.2 | -Mushy-layer thermodynamics with prognostic sea ice salinity;
-EVP;
-ITD with five thickness categories; 3 layers of snow and 8 layers of ice;
-A Delta-Eddington multiple-scattering shortwave radiation treatment, with melt ponds modeled on level, undeformed ice. | Seland et al. (2020) |

When designing the study, the authors limit the number of models considered based on the availability of diagnostic variables: the dynamic and thermodynamic sea ice concentration tendencies. The main result of this choice is to limit the analysis to systems based on different flavors of one sea ice model (CICE). I think including more models might be interesting and more in line with the scope of CMIP6 and OMIP. The tendencies analysis, which is not central to this study, can be limited to the system with the appropriate diagnostic variables.

**Answer:** We have added EC-Earth3 and MIROC6 models for the sea ice concentration (Fig. 1) and ice drift evaluation (Figs. 2 to 4). They are based on the LIM3 and COCO 4.9 sea ice models, respectively. Including

these two models do not change our conclusions on the improved sea ice concentration and ice drift simulations and their connections to the atmospheric forcings. These two models did not provide the dynamic and thermodynamic sea ice concentration tendencies nor the surface fluxes.

**Action:** We have added Figs. 1 to 4 below to the revised manuscript as new Figs. A2, A8-A10 and revised the main text to include these two model results.

[Figure]

**Figure 1. 1980-2007 September and February mean Arctic (a to l) and Antarctic (m to x) sea ice concentration differences between EC-Earth3/C and NSIDC-0051 (first column), EC-Earth3/J and NSIDC-0051 (second column), EC-Earth3/J and EC-Earth3/C (third column), MIROC6/C and NSIDC-0051 (fourth column), MIROC6/J and NSIDC-0051 (fifth column), and MIROC6/J and MIROC6/C (sixth column).**

[Figure]

**Figure 2.** 2003-2007 monthly mean and spatially averaged Arctic ice kinetic energy (MKE) (a to d), ice concentration (e to h), ice thickness (i to l) and surface wind stress (m and n) differences between model/J and model/C. The first to fourth columns correspond to CMCC-CM2-SR5, MRI-ESM2-0, EC-Earth3, MIROC6 model results, respectively. The solid and dashed lines are spatial averages on the regions with ice concentration larger (interior) and smaller (exterior) than 80% in NSIDC-0051, respectively.

[Figure]

**Figure 3. Same as Fig. 2 but for the Antarctic.**

[Figure]

**Figure 4. Differences of significant vector correlation coefficients during 2003–2007 at a level of 99% between model/J and model/C in the Arctic and Antarctic. The first, second, fourth and fifth columns are significant vector correlation coefficients between modeled ice drift and KIMURA/ICDC-NSIDCv4.1 data, and the third column are significant vector correlation coefficients between modeled (CMCC-CM2-SR5 and MRI-ESM2-0) ice drift and surface stress.**

The paper lacks a direct comparison of the reanalysis fields. Also, the CORE2 and JRA55-do forcings have both been bias-corrected in the Arctic to avoid unrealistic model behaviors (e.g. too little sea ice in summer). The correction follows the work of Large and Yeager (2009). How does this bias correction impact your results? To what extent are the forcing converging?

**Answer:** Thanks for your suggestion. The surface air temperature, specific humidity, downward shortwave and longwave radiation fluxes during melting months, and wind speed during freezing months in COREII and JRA55-do are shown in Fig. 5 below. These months are shown because in general the ice concentration simulations are improved from OMIP1 to OMIP2 in summer due to surface heat flux changes and in winter due to the wind stress changes. More information can be found in Figs. 1 to 4, A1 to A4 of our previous manuscript. Compared to COREII, the downward shortwave radiation flux and specific humidity in JRA5-do in the central Arctic Ocean and the coastal region of the western Weddell Sea (Figs. 5g, h, q, r) are

smaller, the downward shortwave radiation flux in the Canadian Arctic Archipelago (CAA) and central Weddell Sea (CWS) regions and the air temperature in the CAA region are larger (Figs. 5h, r, f), and the surface wind speed on Antarctic sea ice in the inner part of the exterior region from 70° to 180°E is weaker (Figs. 5t). These differences in the atmospheric forcing are transferred to the modeled surface fluxes and contribute to the improved ice concentration simulation in those regions. Compared to OMIP1 simulations, the downward shortwave radiation flux and latent heat flux in OMIP2 in the central Arctic Ocean and the coastal region of the western Weddell Sea are smaller, the downward shortwave radiation flux in the CAA and CWS regions and the sensible heat flux in the CAA region are larger, and the surface wind stress on Antarctic sea ice in the inner part of the exterior region from 70° to 180°E is weaker (Figs. 4, 5, A4, A5, A6 in the previous manuscript).

The bias-corrected processes are done in both atmospheric forcings (Tsujino et al., 2018). We can find the JRA55-do forcing is much improved in the Arctic downward shortwave radiation fluxes and this improvement is transferred to the OMIP2 and contribute to an improved ice concentration simulation.

**Action:** We have added Fig. 5 below to the revised manuscript as a new Fig. 6 and added the explanation before on how the differences in the atmospheric forcings are transferred to the model results in the main text.

[Figure]

**Figure 5.** 1980-2007 March-August mean Arctic and October-January mean Antarctic surface air temperature (first column) and specific humidity (second column), downward shortwave (third column) and longwave radiation fluxes (fourth column), as well as October-January mean Arctic and March-August mean Antarctic surface wind speed (fifth column). The first and third rows correspond to COREII, and the second and fourth rows are differences between JRA55-do and COREII.

I believe more observations should be included in the analysis to allow a correct interpretation of the results. We know that in the Arctic different observational datasets are not always in agreement with each other and that identifying the best product is not obvious. I am surprised that this has not been done given that SITool includes multiple observational datasets for sea ice concentration, thickness, and drift. Illustrating the differences between reanalysis in relation to different observational products would certainly be an interesting addition to the study.

**Answer:** Thanks for pointing this out. Two observational datasets are included for the sea ice concentration (NSIDC-0051 and OSI-450), ice drift (KIMURA and ICDC-NSIDCv4.1) and thickness (Envisat and Icesat) evaluations (Figs. 6 to 9) in the revised manuscript. For the ice concentration and ice drift, observational uncertainties are small compared to the model biases. The conclusions on the improved ice concentration and ice drift simulations in OMIP2 do not change by comparing to different observational products (Figs. 6, 7 and 9). The ice thickness observations during 2003-2007 are restricted to a few months per year in both

Envisat and ICESat datasets (Figs. 7 and 8). The Envisat data includes ice thickness from November to April for the Arctic with coverage up to 81.5$^o$N and May to October for the Antarctic from 2003. The ICESat data includes 13 measurement campaigns for the Arctic and 11 for the Antarctic during 2003–2007, and these campaign periods are limited to the months of February–March, March–April, May–June, and October–November, with each campaign lasting roughly 33 d. The comparisons between individual models and the two observational references are thus restricted to these months when data are available. In general, the modeled sea ice thickness is close to the ICESat dataset, while modeled ice thickness is too thick in the Arctic and too thin in the Antarctic compared to Envisat data.

**Action:** These figures are added as new Figs. 1, 7, 8 and 9 in the revised manuscript. We have added these data information and the comparison results in the main text.

[Figure]

**Figure 6. 1980-2007 September and February mean Arctic (a to j) and Antarctic (k to t) sea ice concentration from the NSIDC-0051 data (first column), differences between OSI-450 and NSIDC-0051 (second column), NorESM2-LM/C and NSIDC-0051 (third column), NorESM2-LM/J and NSIDC-0051 (fourth column), and NorESM2-LM/J and NorESM2-LM/C (fifth column). The black lines are contours of 80% concentration (a, f, k and p), which delineate the interior and exterior domains to compute spatial averages in Table 1.**

[Figure]

**Figure 7. 2003-2007 monthly mean and spatially averaged Arctic ice kinetic energy (MKE) (a), ice concentration (c), ice thickness (e) and surface wind stress (g) from observations (blue and purple), NorESM2-LM/C (orange) and NorESM2-LM/J (green). Two observational datasets are included for ice MKE (KIMURA and ICDC-NSIDCv4.1), concentration (NSIDC-0051 and OSI-450) and thickness (Envisat and Icesat). The Envisat ice thickness data is provided from November to April and the coverage is limited up to 81.5°N. The measurement campaigns of Icesat ice thickness is for the months of February–March, March–April, May–June, and October–November, with each campaign lasting roughly 33 d. The solid and dashed lines are spatial averages on the regions with ice concentration larger (interior) and smaller (exterior) than 80% in NSIDC-0051, respectively. The differences between NorESM2-LM/J and NorESM2-LM/C ice MKE (b), ice concentration (d), ice thickness (f) and surface wind stress (h) in the interior (black solid) and exterior (dashed) regions are shown.**

[Figure]

**Figure 8. Same as Fig. 7 but for the Antarctic. The Envisat ice thickness data is provided from May to October.**

[Figure]

**Figure 9. The significant vector correlation coefficients during 2003–2007 at a level of 99% between modeled ice drift (NorESM2-LM/C) and two observational data (KIMURA and ICDC-NSIDCv4.1), respectively, and between NorESM2-LM/C modeled ice drift and surface wind stress in the Arctic (a to c) and Antarctic (g to i). The second and fourth rows are the vector correlation coefficient differences by changing the modeled ice drift from NorESM2-LM/C to NorESM2-LM/J.**

FIGURES AND TABLES:

Tables 1 and 2: I find this table very hard to read and overcrowded. It might be a personal preference, but I could read these data more easily when converted into plots and reorganized. In particular, the use of parenthesis and bold and italic font is confusing. I suggest fully rethinking this, and possibly working with colors/symbols instead of font styles.

Thanks for your comments. We have modified the tables and used different colors to mark them.

Figure 1 and beyond: Showing the results of just one model is, in my opinion, limiting. I understand the authors are concerned about having too many display items in the manuscript but storing relevant material in the appendix is not necessarily a solution. If the space is a concern, why not report only the maps with the difference between the two forcings in the main text, while moving the bias to the appendix? Also, the addition of the observed sea ice concentration in Fig 1 is not very insightful, or at least not a priority for the panel. The same is true in the following figures. Again, this is a suggestion and I realize it depends on personal preferences.

Thanks for your comment. We prefer to include the bias with respect to the observations in the figures. By providing the observational ice concentration and the bias with respect to the observations, we can identify where are the errors (positive and negative values). Then, the differences between OMIP2 and OMIP1 can inform us where are the improvements.

Figures 6 and 7: The color choice of the bar plots is very unhappy. Please consider differentiating the colors of the interior vs. exterior bars.

Thanks for pointing this out. We have changed the bar plots to lines and the interior and exterior regions are shown in solid and dashed lines, respectively, as shown in Figs. 7 and 8.

Yours sincerely,

Xia Lin, François Massonnet, Thierry Fichefet, Martin Vancoppenolle

---

## Author Response (AR1)

Dear reviewer,

We thank you for the constructive comments on the earlier version of the manuscript. We have revised our manuscript following the comments and our response is as follows.

**Reviewer 1:**

The manuscript "Impact of atmospheric forcing uncertainties on Arctic and Antarctic sea ice simulation in CMIP6 OMIP" by Lin et al. 2022 presents a study, that addresses the impact of different atmospheric forcing on the CMIP6 Ocean Model Intercomparison Project (OMIP). The paper focuses on the simulated sea ice condition by three OMIP models that provide sea ice tendencies for a complete analysis of the results.

The study is well structured and organized. It addresses the newest model versions of model intercomparison project, the CMIP6. Methods and data are appropriate, and the results are clearly presented and described (text and figures). The referencing is good. The paper presents a useful report of the model experiments with different forcing datasets, but could profit from more ambitious interpretation of the physical processing behind the results, and evaluation of the atmospheric results against reference datasets.

Thanks for your time and positive evaluation.

Here I suggest three main points that the authors could consider to further improve the manuscript:
- Comparison of the forcing datasets (COREII and JRA55-do)
  - You could compare the surface energy fluxes and the near surface wind speeds of the two input datasets. This would allow to say how directly the differences in the input data are transferred to the model results

**Answer:** Thanks for your suggestion. The surface air temperature, specific humidity, downward shortwave and longwave radiation fluxes during melting months, and wind speed during freezing months in COREII and JRA55-do are shown in Fig. 1 below. The selection to show these variables during melting/freezing months is because in general the ice concentration simulations are improved from OMIP1 to OMIP2 in summer due to surface heat flux changes and in winter due to the wind stress changes. Compared to CORE-II, the downward shortwave radiation flux and specific humidity in the central Arctic Ocean (Figs. 1g and h) and specific humidity the coastal region of the western Weddell Sea (Fig. 1q) in JRA5-do are smaller, the downward shortwave radiation flux in the CAA and CWS regions (Figs. 1h and r) and the air temperature in the CAA region are larger (Fig. 1f), and the surface wind speed on Antarctic sea ice in the inner part of the exterior region from 70° to 180°E is weaker (Fig. 1t). These differences in the atmospheric forcing are transferred to the surface fluxes and contribute to the improved ice concentration simulation in

those regions. Compared to OMIP1 simulations, the downward shortwave radiation flux and latent heat flux in the central Arctic Ocean and latent heat flux in the coastal region of the western Weddell Sea in OMIP2 are smaller, the downward shortwave radiation flux in the CAA and CWS regions and the sensible heat flux in the CAA region are larger (Figs. 5, A6, A7, Table 4 in the revised manuscript), and the surface wind stress on Antarctic sea ice in the inner part of the exterior region from 70° to 180°E is weaker (Figs. 4, A5, Table 3 in the revised manuscript).

**Action:** We have added Fig. 1 below to the revised manuscript as a new Fig. 6 and added the explanation on how the differences in the atmospheric forcings are transferred to the model results in lines 441-458.

[Figure]

**Figure 1. 1980-2007 March-August mean Arctic and October-January mean Antarctic surface air temperature (first column) and specific humidity (second column), downward shortwave (third column) and longwave radiation fluxes (fourth column), as well as October-January mean Arctic and March-August mean Antarctic surface wind speed (fifth column). The first and third rows correspond to COREII, and the second and fourth rows are differences between JRA55-do and COREII.**

- More focus on the evaluation and interpretation of the energy balance terms

o Are the model results realistic, are OMIP2 results better that OMIP1? You could compare the modeled fluxes against literature values or reference data. I understand that finding a good reference dataset for polar energy fluxes is hard. Still, it would be possible to compare the fluxes against reanalyses (eg. ERA5, JRA55 or NCEP-CFSR) to get an idea of the possible biases. Another approach would be to compare each model against the multimodel mean (mean of CMCC-CM2-SR5, MRI-ESM2-0 and NorESM2-LM), assuming that the multimodel mean is better than each individual model.

**Answer:** We compared the simulated surface fluxes to ERA5 reanalysis data. The multi-model mean includes each model result, so comparing a model to it is not relevant if we want to study the skill. The 12-hourly ERA5 data of sea ice concentration, surface sensible, latent, net shortwave and longwave radiation fluxes, northward and eastward surface stresses on Earth surface from 1980 to 2007 are used to calculate the surface fluxes on sea ice. The net surface heat fluxes during melting months and the surface stress during freezing months (Fig. 2 below), as well as surface sensible, latent heat fluxes, net shortwave and longwave radiation fluxes during melting months (Fig. 3 below) from ERA5 and the differences between NorESM2-LM/C and ERA5, and between NorESM2-LM/J and NorESM2-LM/C are shown below.

As introduced in section 3.1.2, compared to NorESM2-LM/C simulations, the downward net surface heat fluxes during melting months in NorESM2-LM/J are smaller in the central Arctic Ocean and over the coastal regions of the western Weddell Sea, and larger in the CAA and CWS regions. The changed downward net surface heat flux changes in NorESM2-LM/J compared to NorESM2-LM/C contribute to the improved September ice concentration simulation in those regions. It can be observed in Figs. 2a to h that the downward net surface heat flux in NorESM2-LM/J in these regions is close to the ERA5 reanalysis data. The contributions from the surface sensible, latent heat fluxes and the net shortwave and longwave radiations to the improvements are explained in section 3.1.2. These surface fluxes in NorESM2-LM/J in these regions are close to the ERA5 reanalysis data (Fig. 3).

Compared to NorESM2-LM/C simulations, the decreased surface wind stress in NorESM2-LM/J in the inner part of the exterior region from 70° to 180°E (Fig. 2p) slows down the ice motion and improve the modeled September ice concentration in the exterior region from 70° to 180°E. The surface wind stress in NorESM2-LM/J in these regions is close to the ERA5 reanalysis data (Figs. 2m to p).

Even though the NorESM2-LM/J simulations are close to ERA5 reanalysis data in some regions, we found that there are large surface flux differences between model outputs and ERA5 data. It is very hard to qualify surface energy balance fluxes and observational uncertainties are large. Graham et al. (2019) evaluated six atmospheric reanalyses over Arctic sea ice from winter to early summer and found that the surface heat fluxes bias in reanalyses are large in general, in particular turbulent fluxes, for instance.

Compared to OMIP1, the OMIP2 simulations on surface fluxes can be considered better because the JRA55-do atmospheric forcing is relatively new and with higher temporal and horizontal resolution compared to CORE-II, as detailed in Tsujino et al. (2018).

**Action:** We prefer not to include ERA5 data in the revised paper due to the large uncertainties in the surface fluxes. However, we indeed have added these contents in lines 357-360, 427-428, and 613-615 of the revised manuscript.

[Figure]

**Figure 2. 1980-2007 March-August mean Arctic net surface heat flux (a to d) and Antarctic surface stress (m to p), and October-January mean Antarctic net surface heat flux (e to h) and Arctic surface stress (i to l). The positive values indicate a surface flux downward. The first column corresponds to ERA5 data and the second to fourth columns are differences between NorESM2-LM/C and ERA5, between NorESM2-LM/J and ERA5, and between NorESM2-LM/J and NorESM2-LM/C, respectively.**

[Figure]

**Figure 3.** 1980-2007 March-August mean Arctic (a to l) and October-January mean Antarctic (m to x) surface sensible (first column) and latent heat fluxes (second column), net shortwave (third column) and longwave radiation fluxes (fourth column). The positive values indicate a heat flux downward. The first and fourth rows correspond to ERA5 data, and the second and fifth rows are differences between NorESM2-LM/C and ERA5, and the third and sixth rows are differences between NorESM2-LM/J and NorESM2-LM/C.

- Evaluation of the sea ice thickness
    - There are substantial biases in the sea ice thickness. Can you comment them in any way? Is the reference data reliable (see also my comment for Page 17, lines 399-400), can the limitations in model physics make the ice too thick, what in the forcing data or initial conditions could cause the ice to be too thick?

**Answer:** The ice thickness observations during 2003-2007 are restricted to a few months per year in both Envisat and ICESat datasets. The Envisat data includes ice thickness from November to April for the Arctic with coverage up to 81.5°N and May to October for the Antarctic from 2003. The ICESat data includes 13 measurement campaigns for the Arctic and 11 for the Antarctic during 2003–2007, and these campaign periods are limited to the months of February–March, March–April, May–June, and October–November, with each campaign lasting roughly 33 d. The comparisons between individual models and the two observational references are thus restricted to these months when data are available.

The estimated ICESat sea ice thickness is added in Fig. 4 below. In general, modeled sea ice thickness is close to sea ice thickness in the ICESat dataset, while modeled ice thickness is too thick in the Arctic and too thin in the Antarctic compared to Envisat data. We average modeled ice thickness limited up to 81.5°N and this affects the ice thickness in the summer months but not from November to April. There is no consistent improvement of the representation of sea ice thickness in both hemispheres by changing the atmospheric forcing from COREII to JRA55-do (Fig. 4). More observations and studies are needed to evaluate the ice thickness bias in models.

**Action:** We have added ICESat sea ice thickness to the revised manuscript as new Figs. 7 and 8, and added the contents in lines 486-494 and 516-519.

[Figure]

Figure 4. 2003-2007 monthly mean and spatially averaged Arctic (a) and Antarctic (b) ice thickness from Envisat (blue marks), ICESat (purple marks), NorESM2-LM/C (orange) and NorESM2-LM/J (green). The Envisat ice thickness data is provided from November to April and the coverage is limited up to 81.5°N. The measurement campaigns of ICESat ice thickness is for the months of February–March, March–April, May–June, and October–November, with each campaign lasting roughly 33 d. The solid and dashed lines are spatial averages on the regions with ice concentration larger (interior) and smaller (exterior) than 80% in NSIDC-0051, respectively. The light orange and green lines in the Arctic (a) are modeled ice thickness averaged limited up to 81.5°N.

These improvements could make the paper even more relevant to the audience and turn the "clues on how improved atmospheric reanalysis products influence sea ice simulations" to more than just clues. However, even in the current form, the paper points out the aspects where more research is needed, which is also very relevant for the community. All in all, this is a meticulously prepared manuscript, where the expertise of the authors is visible. Thank you for the interesting reading!

Thanks a lot.

I provide some more detailed comments below.

Page 3, Line 87: In this paragraph, you could summarize some main links between the atmospheric circulation and sea ice conditions. You could describe the role of atmospheric circulation and it's impacts on sea ice on a general level. You could mention that some circulation patters allow heat and moisture transport to the Arctic, while others prohibit it.

Thanks for your suggestion. We have added more details in lines 95-102 of the revised manuscript. The spatial variability of sea ice concentration and its links with the atmospheric circulation vary with season. The change in the position and strength of the cyclonic or anticyclonic circulation over the sea ice can affect the sea ice motion and freezing/melting (Rigor et al., 2002; Raphael and Hobbs, 2014; Ding et al., 2017). Strong winter wind-driven ice exports in the Eurasian coastal region occur during high North Atlantic Oscillation (NAO) index years, which can contribute to the reduction of summer Arctic sea ice extent observed during the 1980s and 1990s (Hu et al., 2002). In the Antarctic, the decreases of sea ice concentration generally occur in regions of poleward flow and the increases of sea ice concentration occur in regions of equatorward flow (Renwick et al., 2012).

Page 5, Lines 165-166: To help the reader, you could mention that SIC was overestimated in CCA in OMIP1 and that overestimation was reduced in OMIP2.

We have added the information in lines 204-206.

Page 8, Table 1: The caption states: "The improvements on ice concentration simulations and the related reasons in summer (bold) and winter (bold italic) are marked". Bold and italic text is used also for the sea ice concentration tendencies, the surface heat flux, and the surface stress on sea ice. Is this correct, or should the improvements be marked only for the SIC?

Yes, the related sea ice concentration tendencies, the surface heat flux and the surface stress on sea ice are also marked. We have rephrased the sentence in the caption of Tables 2 and 3.

Page 13, line 319: You wrote "downward net shortwave radiation flux" do you mean the net shortwave or the downward shorwave?

Thanks for pointing this out. It is 'net shortwave radiation flux'. The text has been clarified.

Page 16, Table 2: To me, it would be more intuitive to see the OMIP1 (C) and OMIP2 (J) values instead of C and J-C. Best would be to have all three values (C, J, and J-C), but it's hard to fit them in one table. Also, I would suggest adding the downward longwave and upward longwave radiation, as you did for the shortwave radiation. You will see if the decreased downward shortwave radiation coincides with increased downward longwave radiation. This would suggest an increase in atmospheric moisture/cloudiness between the two experiments.

Thanks for your suggestion. We have modified them to OMIP1 [C] and OMIP2 [J] values in Table 4 of the revised manuscript. The downward and upward longwave radiation fluxes are shown in Fig. 5 below and the values are added in Table 4. The decreased downward shortwave radiation in the Arctic interior region does not coincide with increased downward longwave radiation. The connections between downward shortwave and longwave radiation fluxes are not direct. Both temperature and humidity can affect the downward longwave radiation flux.

[Figure]

Figure 5. 1980-2007 March-August mean Arctic (a to h) and October-January mean Antarctic (i to p) downward and upward longwave radiation fluxes in NorESM2-LM (first two columns) and CMCC-CM2-SR5 (last two columns). The first and third rows correspond to model/C, and the second and fourth rows are differences between model/J and model/C. The positive values indicate a surface heat flux downward.

Page 17, line 390: Could you add a definition or equation for the MKE?

Yes, we have added the equation of the MKE in lines 471-473.

Page 17, lines 399-400: Could you select the modeled sea ice thickness for a domain that matches the Envisat data (< 81.5°N)? This would allow a better comparison between the datasets.

We have added this calculation for a better comparison. This is explained in the response to the third main point before.

Page 17, line 402: Should "ice drift speed" be "ice-motion MKE"?

Thanks for pointing this out. We now use 'ice-motion MKE' in the revised manuscript.

Page 18, Figure 6: The interior-bar in June is all green. I think that the blue and orange bars might be missing/hidden.

We have changed bars to lines in new Figs. 7 and 8 to make it clear.

Page 19, line 434: "drift speeds" or "MKE"?

We have checked this sentence in the full text and modified it.

Page 22, line 515: "More attention needs to be paid to the radiation fluxes and wind stress in the atmospheric reanalysis products." Here you could elaborate on what kind of attention is needed. What are the sources of the current problems, what caused the improvements when passing from CORE-II to JRA55-do etc.

Thanks for your comment. We have explained the reasons for improvements when passing from CORE-II to JRA55-do in the manuscript. Some aspects of the sea ice simulation are not improved by changing the forcing from CORE-II to JRA55-do, such as the winter Arctic ice concentration in the exterior region, summer Antarctic ice concentration in the coastal regions and Antarctic ice drift speed. The bias in surface heat fluxes and surface stress in these regions are large in NorESM-LM/J compared to ERA5 (Fig. 2). Improving Antarctic radiation fluxes, and Arctic and Antarctic winds in the atmospheric reanalysis products can be helpful to reduce the bias. We have added these comments in lines 611-616 of the discussion section.

Yours sincerely,

Xia Lin, François Massonnet, Thierry Fichefet, Martin Vancoppenolle

Dear reviewer,

We thank you for the constructive comments on the earlier version of the manuscript. We have revised our manuscript following the comments and our response is as follows.

**Reviewer 2:**

First, I apologize to the editor and authors for the delay in writing this review.

The study by Lin et al. investigates an important aspect of stand-alone ocean and sea ice models, which is the relation between the simulated fields and the atmospheric forcings. In simple terms, the paper tries to assess whether a new and arguably better forcing (JRA-55-do) leads to better simulation compared to an older forcing (CORE2). I find this a very interesting question, with important implications also for fully coupled model setups, and I would like to see more studies on these technical but rather important aspects. Congratulations to the authors for pursuing such an interesting problem.

That said, I found the authors' methodological approach unsatisfactory in providing a convincing answer to the problem. The analyses presented here are formally correct, but several central aspects have been almost completely neglected, as illustrated in my comments below. Otherwise, the paper is very well written and structured, but I have some suggestions for improving figures and tables, which sometimes are not adequate.

In summary, I have several major concerns that in my opinion should be addressed before this manuscript is considered for publication. I hope the authors find these helpful for improving their study.

MAJOR COMMENTS:

My biggest concern is that this manuscript does not consider the tuning of the systems analyzed. Let me start by acknowledging that finding out specific details about namelist parameters and other technical information is not trivial when dealing with CMIP-type simulations. Nevertheless, this aspect is important for a study of this kind and cannot be neglected. Specifically, I am wondering whether the CORE2 and JRA-55-do simulations were run with the same model setup, or if the model was specifically tuned for a certain forcing. In my view, tuning is a fundamental step that, given the under-constrained nature and spatiotemporal variability of the sea ice model parameters, must be performed to accommodate a model configuration to a specific forcing. I would argue that the best setup for a study like this would be one where each simulation is optimized to obtain the best compatibility with a set of observations, given a specific atmospheric forcing (i.e., different parameters for CORE2 and JRA-55-do). If this is not the case and an identical model setup is adopted for all atmospheric forcings, we should at least be made aware of whether the model parameters

have been tuned under CORE2 or JRA-55-do, if tuned at all for an OMIP setup. For example, if a model configuration has been tuned under JRA-55 and this same configuration is then run under CORE2, it is not surprising that one would outperform the other. Based on the information currently in the paper, we cannot say anything regarding the previous considerations, which, in my view, is a substantial limitation that should be addressed to pursue the research question from the right angle.

**Answer:** Thanks for your comment. We recognize that tuning is a key aspect in climate models. The design of the CMIP6 OMIP simulations has been organized by the World Climate Research Programme (WCRP) Climate Variability and Predictability (CLIVAR) Working Group on Ocean Model Development Panel (OMDP), and ongoing research collaboration is done through the OMDP to develop OMIP2 (Griffies et al., 2016). Importantly, and to our understanding, the same configuration is used under two different atmospheric forcing datasets as mentioned in Tsujino et al. (2020). It is possible that some groups could have tuned for OMIP2 and then used the same setup for OMIP1, so part of the improvement with OMIP2 could be due to this experimental setup choice. From our analysis, the differences in the atmospheric forcing are transferred to the modeled surface fluxes and contribute to the improved ice concentration simulation (see our response to major comment 4, Fig. 5 below). Thus, the OMIP1 and OMIP2 datasets allow singling out how atmospheric forcing differences are translated into simulated sea ice differences.

**Action:** The information has been added in lines 81-85 of the revised introduction when introducing the OMIP models. We also added one sentence in lines 629-633 of the discussion part to address the possible effects from tuning.

I think the manuscript lacks an in-depth description of the model components used in the three systems considered, which could be helpful for a more detailed interpretation of the results. The only information available in Lin et. al. 2021 is that two systems employ different versions of CICE as sea ice model components (CMCC-CM2 and NorESM2). By digging in the MRI-ESM2 model description paper we discover the sea ice component of MRI. COM4.4 is also based on CICE. Given the modularity of CICE, the model version tells us nothing about the physical configuration used by each modeling center. A better description of the model configurations would allow linking differences in the model response to the reanalyses, to differences in the specific physics used. For example, I suspect that different radiative schemes lead to very different sea ice concentration conditions in summer.

**Answer:** Thanks for your suggestion. The details of five sea ice models are provided in Table 1 below. We have added the EC-Earth3 and the MIROC6 models in the revised manuscript (see our response to the next comment) and they are based on the LIM3 and COCO4.9 sea ice models, respectively. In Table 1, we can find the differences in sea ice models. The CMCC-CM2-SR5 and NorESM2-LM models use different versions of CICE sea ice model with different thermodynamics. The sea ice component of MRI-ESM2-0 use

default CICE CCSM3 radiation scheme, but thermodynamics and the melt ponds are different from CICE model. The radiative schemes in EC-Earth3 and NorESM2-LM are different, which can affect the summer sea ice concentration conditions in EC-Earth3 (Fig. 1 below) and NorESM2-LM (Fig. 6 below).

**Action:** These contents have been added in the revised Table 1 and in lines 621-624. We have added a sentence in lines 626-627 of the revised manuscript recommending that, in such inter-comparison exercises, all the specificities/namelists of the sea ice models used should systematically be reported to help understand the different responses to model physics.

Table 1. The details of five CMIP6-OMIP sea ice models evaluated in the study. Some information can be found on the following link: https://www.cen.uni-hamburg.de/en/icdc/data/cryosphere/cmip6-sea-ice-area.html (last access: 28 November 2022).

| Model | Sea Ice Model | Sea Ice Component | References |
|---|---|---|---|
| CMCC-CM2-SR5 | CICE4 | -Energy-conserving thermodynamics on 1 layer of snow and 4 layers of ice;
 -Elastic-Viscous-Plastic (EVP) rheology;
 -Ice Thickness Distribution (ITD) with 5 thickness categories;
 -A Delta-Eddington multiple-scattering shortwave radiation treatment;
 -Explicit level-ice melt ponds parameterization; | Hunke and Lipscomb (2008);

 Cherchi et al. (2019); |
| EC-Earth3 | LIM3 | -Energy-conserving halo-thermodynamics with prognostic sea ice salinity on 1 layer of snow and 2 layers of ice;
 -EVP;
 -ITD with 5 thickness categories;
 -Empirical albedo function, exponential attenuation of solar radiation in sea ice if no snow;
 -Melt ponds: step reduction in albedo when Tsu = 0°C; | Rousset et al. (2015);
 Döscher et al. (2022); |
| MIROC6 | COCO4.9 | -Energy-conserving thermodynamics on 0 layer of snow (without heat capacity) and 1 layer of ice;
 -EVP;
 -ITD with 5 thickness categories;
 -Empirical albedo function;
 -Implicit melt ponds; | Komuro et al. (2012);

 Tatebe et al. (2019); |
| MRI-ESM2-0 | MRI.COM4.4 | -Energy-conserving thermodynamics following Mellor and Kantha (1989) on 0 layer of snow and 1 layer of ice;
 -EVP;
 -ITD with 5 thickness categories;
 -The "default" CICE CCSM3 radiation scheme;
 -Implicit melt ponds: adjust the albedo based on surface conditions; | Tsujino et al. (2010);

 Hunke et al. (2015);

 Yukimoto et al. (2019); |
| NorESM2-LM | CICE5.1.2 | -Mushy-layer thermodynamics with prognostic sea ice salinity on 3 layers of snow and 8 layers of ice;
 -EVP;
 -ITD with 5 thickness categories;
 -A Delta-Eddington multiple-scattering shortwave radiation treatment;
 -Explicit level-ice melt ponds parameterization. | Hunke et al. (2015);

 Seland et al. (2020). |

When designing the study, the authors limit the number of models considered based on the availability of diagnostic variables: the dynamic and thermodynamic sea ice concentration tendencies. The main result of this choice is to limit the analysis to systems based on different flavors of one sea ice model (CICE). I think including more models might be interesting and more in line with the scope of CMIP6 and OMIP. The tendencies analysis, which is not central to this study, can be limited to the system with the appropriate diagnostic variables.

**Answer:** The EC-Earth3 and MIROC6 models for the sea ice concentration (Fig. 1) and ice drift evaluation (Figs. 2 to 4) are included. They are based on the LIM3 and COCO 4.9 sea ice models, respectively. Including these two models do not change our conclusions on the improved sea ice concentration and ice drift simulations and their connections to the atmospheric forcings. These two models did not provide the dynamic and thermodynamic sea ice concentration tendencies nor the surface fluxes.

**Action:** We have added Figs. 1 to 4 to the revised manuscript as new Figs. A2, A8-A10, included the values from two models in Tables 2 and 3, and revised the full text to include the results of these two models.

[Figure]

**Figure 1. 1980-2007 September and February mean Arctic (a to l) and Antarctic (m to x) sea ice concentration differences between EC-Earth3/C and NSIDC-0051 (first column), EC-Earth3/J and NSIDC-0051 (second column), EC-Earth3/J and EC-Earth3/C (third column), MIROC6/C and NSIDC-0051 (fourth column), MIROC6/J and NSIDC-0051 (fifth column), and MIROC6/J and MIROC6/C (sixth column).**

[Figure]

**Figure 2. 2003-2007 monthly mean and spatially averaged Arctic ice kinetic energy (MKE) (a to d), ice concentration (e to h), ice thickness (i to l) and surface wind stress (m and n) in model/C (orange), model/J (green), and differences between model/J and model/C (black). The first to fourth columns correspond to CMCC-CM2-SR5, MRI-ESM2-0, EC-Earth3, MIROC6 model results, respectively. The solid and dashed lines are spatial averages on the regions with ice concentration larger (interior) and smaller (exterior) than 80% in NSIDC-0051, respectively.**

[Figure]

**Figure 3. Same as Fig. 2 but for the Antarctic.**

[Figure]

**Figure 4. Differences of significant vector correlation coefficients during 2003–2007 at a level of 99% between model/J and model/C in the Arctic (a to j) and Antarctic (k to t). The first, second, fourth and fifth columns are significant vector correlation coefficients between modeled ice drift and KIMURA/ICDC-NSIDCv4.1 data, and the third column are significant vector correlation coefficients between modeled (CMCC-CM2-SR5 and MRI-ESM2-0) ice drift and surface stress.**

The paper lacks a direct comparison of the reanalysis fields. Also, the CORE2 and JRA55-do forcings have both been bias-corrected in the Arctic to avoid unrealistic model behaviors (e.g. too little sea ice in summer). The correction follows the work of Large and Yeager (2009). How does this bias correction impact your results? To what extent are the forcing converging?

**Answer:** Thanks for your suggestion. The surface air temperature, specific humidity, downward shortwave and longwave radiation fluxes during melting months, and wind speed during freezing months in COREII and JRA55-do are shown in Fig. 5 below. The selection to show these variables during melting/freezing months is because in general the ice concentration simulations are improved from OMIP1 to OMIP2 in summer due to surface heat flux changes and in winter due to the wind stress changes. Compared to CORE-II, the downward shortwave radiation flux and specific humidity in the central Arctic Ocean (Figs.

5g and h) and specific humidity the coastal region of the western Weddell Sea (Fig. 5q) in JRA5-do are smaller, the downward shortwave radiation flux in the CAA and CWS regions (Figs. 5h and r) and the air temperature in the CAA region are larger (Fig. 5f), and the surface wind speed on Antarctic sea ice in the inner part of the exterior region from 70° to 180°E is weaker (Fig. 5t). These differences in the atmospheric forcing are transferred to the surface fluxes and contribute to the improved ice concentration simulation in those regions. Compared to OMIP1 simulations, the downward shortwave radiation flux and latent heat flux in the central Arctic Ocean and latent heat flux in the coastal region of the western Weddell Sea in OMIP2 are smaller, the downward shortwave radiation flux in the CAA and CWS regions and the sensible heat flux in the CAA region are larger (Figs. 5, A6, A7, Table 4 in the revised manuscript), and the surface wind stress on Antarctic sea ice in the inner part of the exterior region from 70° to 180°E is weaker (Figs. 4, A5, Table 3 in the revised manuscript).

The bias-corrected processes are done in both atmospheric forcings (Tsujino et al., 2018). We can find the JRA55-do forcing is much improved in the Arctic downward shortwave radiation fluxes and this improvement is transferred to the OMIP2 and contribute to an improved ice concentration simulation.

**Action:** We have added Fig. 5 below to the revised manuscript as a new Fig. 6 and added the explanation on how the differences in the atmospheric forcings are transferred to the model results in lines 441-458.

[Figure]

**Figure 5. 1980-2007 March-August mean Arctic and October-January mean Antarctic surface air temperature (first column) and specific humidity (second column), downward shortwave (third column) and longwave radiation fluxes (fourth column), as well as October-January mean Arctic and March-August mean Antarctic surface wind speed (fifth column). The first and third rows correspond to COREII, and the second and fourth rows are differences between JRA55-do and COREII.**

I believe more observations should be included in the analysis to allow a correct interpretation of the results. We know that in the Arctic different observational datasets are not always in agreement with each other and that identifying the best product is not obvious. I am surprised that this has not been done given that SITool includes multiple observational datasets for sea ice concentration, thickness, and drift. Illustrating the differences between reanalysis in relation to different observational products would certainly be an interesting addition to the study.

**Answer:** Thanks for pointing this out. Two observational datasets are included for the sea ice concentration (NSIDC-0051 and OSI-450), ice drift (KIMURA and ICDC-NSIDCv4.1) and thickness (Envisat and Icesat) evaluations (Figs. 6 to 9). For the ice concentration and ice drift, observational uncertainties are small

compared to the model biases. The conclusions on the improved ice concentration and ice drift simulations in OMIP2 do not change by comparing to different observational products. The ice thickness observations during 2003-2007 are restricted to a few months per year in both Envisat and ICESat datasets (Figs. 7 and 8). The Envisat data includes ice thickness from November to April for the Arctic with coverage up to 81.5$^{o}$N and May to October for the Antarctic from 2003. The ICESat data includes 13 measurement campaigns for the Arctic and 11 for the Antarctic during 2003–2007, and these campaign periods are limited to the months of February–March, March–April, May–June, and October–November, with each campaign lasting roughly 33 d. The comparisons between individual models and the two observational references are thus restricted to these months when data are available. In general, the modeled sea ice thickness is close to the ICESat dataset, while modeled ice thickness is too thick in the Arctic and too thin in the Antarctic compared to Envisat data.

**Action:** These figures are added as new Figs. 1, 7, 8 and 9 in the revised manuscript. We have added these data information in lines 156-166 and the comparison results in the main text.

[Figure]

**Figure 6. 1980-2007 September and February mean Arctic (a to j) and Antarctic (k to t) sea ice concentration from the NSIDC-0051 data (first column), differences between OSI-450 and NSIDC-0051 (second column), NorESM2-LM/C and NSIDC-0051 (third column), NorESM2-LM/J and NSIDC-0051 (fourth column), and NorESM2-LM/J and NorESM2-LM/C (fifth column). The black lines are contours of 80% concentration (a, f, k and p), which delineate the interior and exterior domains to compute spatial averages in Tables 2 to 4.**

[Figure]

**Figure 7. 2003-2007 monthly mean and spatially averaged Arctic ice kinetic energy (MKE) (a), ice concentration (c), ice thickness (e) and surface wind stress (g) from observations (blue and purple), NorESM2-LM/C (orange) and NorESM2-LM/J (green). Two observational datasets are included for ice MKE (KIMURA and ICDC-NSIDCv4.1), concentration (NSIDC-0051 and OSI-450) and thickness (Envisat and Icesat). The solid and dashed lines are spatial averages on the regions with ice concentration larger (interior) and smaller (exterior) than 80% in NSIDC-0051, respectively. The differences between NorESM2-LM/J and NorESM2-LM/C ice MKE (b), ice concentration (d), ice thickness (f) and surface wind stress (h) in the interior (black solid) and exterior (dashed) regions are shown.**

[Figure]

**Figure 8. Same as Fig. 7 but for the Antarctic. The Envisat ice thickness data is provided from May to October.**

[Figure]

**Figure 9. The significant vector correlation coefficients during 2003–2007 at a level of 99% between modeled ice drift (NorESM2-LM/C) and two observational data (KIMURA and ICDC-NSIDCv4.1), respectively, and between NorESM2-LM/C modeled ice drift and surface wind stress in the Arctic (a to c) and Antarctic (g to i). The second and fourth rows are the vector correlation coefficient differences by changing the modeled ice drift from NorESM2-LM/C to NorESM2-LM/J.**

FIGURES AND TABLES:

Tables 1 and 2: I find this table very hard to read and overcrowded. It might be a personal preference, but I could read these data more easily when converted into plots and reorganized. In particular, the use of parenthesis and bold and italic font is confusing. I suggest fully rethinking this, and possibly working with colors/symbols instead of font styles.

Thanks for your comments. We have split original Table 1 to Tables 2 and 3 in the revised manuscript and used different colors to mark them.

Figure 1 and beyond: Showing the results of just one model is, in my opinion, limiting. I understand the authors are concerned about having too many display items in the manuscript but storing relevant material in the appendix is not necessarily a solution. If the space is a concern, why not report only the maps with the difference between the two forcings in the main text, while moving the bias to the appendix? Also, the addition of the observed sea ice concentration in Fig 1 is not very insightful, or at least not a priority for the panel. The same is true in the following figures. Again, this is a suggestion and I realize it depends on personal preferences.

Thanks for your comment. We prefer to include the bias with respect to the observations in the figures. By providing the observational ice concentration and the bias with respect to the observations, we can identify where are the errors (positive and negative values). Then, the differences between OMIP2 and OMIP1 can inform us where are the improvements.

Figures 6 and 7: The color choice of the bar plots is very unhappy. Please consider differentiating the colors of the interior vs. exterior bars.

Thanks for pointing this out. We have changed the bar plots to lines and the interior and exterior regions are shown in solid and dashed lines, respectively, as shown in Figs. 7, 8, A8 and A9 in the revised manuscript.

Yours sincerely,
Xia Lin, François Massonnet, Thierry Fichefet, Martin Vancoppenolle

---

## Author Response (AR2)

Dear editor,

Thanks for the constructive comments from you and two reviewers. Minor text issues pointed out by reviewer 1 have been corrected in lines 147, 472 and 475 of the revised manuscript. We have revised our manuscript following the comments from reviewer 2 and our response is as follows.

**Reviewer 2:**

Thank you for the accurate revision of this paper and the answers to my comments. Although a full investigation of the issues I mentioned in my review is in part still missing (impact of model setup and tuning), I am happy to see that these aspects are now discussed in the concluding paragraphs of the paper, which should help the readers to contextualize the message. I still have some minor additional comments that I suggest considering before publishing the paper.

Figure 6: When you calculate the difference between JRA-55 and CORE2 we can see that there are some interpolation issues. I am not sure what is going on there, but I suggest checking your analyses to fix the issue. If this is a true feature of the reanalyses, you should describe it.

Thanks for pointing this out. We have checked the interpolation and updated the Fig. 6 in the manuscript. This is because the resolution of COREII data is $2^o$ and we interpolated them to the $1/4°$ before. Now, we interpolate them to the COREII grid to avoid this issue.

Figures 7 and 8: I am very confused by the behavior of the sea ice thickness in summer for both hemispheres. I would expect a decline in the thickness, which we do not observe. Is this a consequence of the averaging conditioned to the NSIDC sea ice concentration? I think an explanation should be given, or possibly choose a different way of comparing the model with the observations.

**Answer:** Thanks for this constructive comment. A peak ice thickness in August is noticed in NorESM2-LM, CMCC-CM2-SR5, EC-Earth3 and MIROC6 except in MRI-ESM2-0. The regions selected to do the spatial average are different in each month and this can affect the monthly mean ice thickness. Meanwhile, we have compared the monthly mean ice thickness, concentration and vectors between NorESM2-LM/C and MRI-ESM2-0/C (Figs. 1 to 3). The much increased ice thickness in August is related to the much decreased ice concentration and increased dynamic convergence in NorESM2-LM.

[Figure]

Fig. 1 The 2003-2007 monthly mean ice thickness of NorESM2-LM/C and MRI-ESM2-0/C.

[Figure]

Fig. 2 The 2003-2007 monthly mean ice concentration of NorESM2-LM/C and MRI-ESM2-0/C.

[Figure]

Fig. 3 The 2003-2007 monthly mean ice vectors of NorESM2-LM/C and MRI-ESM2-0/C.

**Action:** These discussions are added in lines 521-526 of the revised manuscript. This is not due to our methods of comparison, but we do check the calculation. As introduced in Lin et al. (2021), the ice vectors from observations and models are removed when ice concentrations are below 50 %, or the data are closer than 75 km to the coast, or with a spurious value to reduce the spatial and temporal noise in the calculation. To be consistent, we apply these selections to other variables in the revised Figs. 7, 8 A8 and A9. These are detailed in lines 479-482. We revised the corresponding text in section 3.2.1 and this update did not change the conclusion.

Yours sincerely,

Xia Lin, François Massonnet, Thierry Fichefet, Martin Vancoppenolle